# Stepwise assembly and release of Tc toxins from *Yersinia entomophaga*

Miki Feldmüller [1], Charles F. Ericson [1,5], Pavel Afanasyev [2,5], Yun-Wei Lien [1,5], Gregor L. Weiss [1], Florian Wollweber [1], Marion Schoof [3,4], Mark Hurst [3,4] & Martin Pilhofer [1]✉

Tc toxins are virulence factors of bacterial pathogens. Although their structure and intoxication mechanism are well understood, it remains elusive where this large macromolecular complex is assembled and how it is released. Here we show by an integrative multiscale imaging approach that *Yersinia entomophaga* Tc (YenTc) toxin components are expressed only in a subpopulation of cells that are 'primed' with several other potential virulence factors, including filaments of the protease M66/StcE. A phage-like lysis cassette is required for YenTc release; however, before resulting in complete cell lysis, the lysis cassette generates intermediate 'ghost' cells, which may serve as assembly compartments and become packed with assembled YenTc holotoxins. We hypothesize that this stepwise mechanism evolved to minimize the number of cells that need to be killed. The occurrence of similar lysis cassettes in diverse organisms indicates a conserved mechanism for Tc toxin release that may apply to other extracellular macromolecular machines.

Microbial cell–cell interactions are often mediated by macromolecular complexes, frequently toxins, which have to be translocated into the extracellular space. For most bacterial toxins, this is achieved by classical secretion systems[1,2]. For particularly large macromolecular toxins, however, cells face the challenge of having to either translocate huge complexes across the cell envelope or assemble the complex in the extracellular space. One prime example of a system facing this challenge are tripartite toxin complexes (Tc toxins), which are multicomponent toxins with a molecular weight of >1 MDa[3]. Tc toxins are insecticidal toxins, and homologous genes were identified in pathogens of insects, plants and humans[3–5]. The first Tc toxin gene cluster was identified in *Photorhabdus luminescens*[6] and homologues of Tc components have since been detected in a range of Gram-negative and Gram-positive bacteria[3,7].

Tc toxins typically consist of three subunits whose assembly into a holotoxin is required for delivery and toxicity[8]. Subunit A (composed of five TcA protomers) comprises a central α-helical translocation channel connected to a receptor-binding outer shell[8–10]. The pentamer is required for target cell association, membrane penetration as well as toxin translocation[8,10]. Subunits B and C (TcB/TcC) form a heterodimer, which encapsulates the autoproteolytically cleaved hypervariable region (HVR) of TcC[10–12]. HVR represents the actual cargo, which is a highly potent and non-selective cytotoxic effector. Binding of TcA to TcB–TcC, accompanied by conformational changes, results in the fully assembled and active holotoxin[3,10,12]. The holotoxin binds to target cell surface receptors[13–17] and undergoes endocytosis[3,10]. A pH shift triggers the opening of the TcA shell and a syringe-like translocation of the toxin into the target[10,18]. The toxin then disrupts cellular processes, leading to cell death[3,19,20]. While the structure and intoxication mechanism of Tc toxins are well understood, the site of toxin assembly and the mechanism of release from the bacteria remain unclear. Here we study Tc toxin assembly and release from *Yersinia entomophaga* MH96 (ref. 21), hereafter called *Y. entomophaga* wild type (WT). We point out a recent preprint on the regulation and release of Tc toxin from this organism, which is complementary to the data shown here[22].

[1]Department of Biology, Institute of Molecular Biology and Biophysics, Eidgenössische Technische Hochschule Zürich, Zürich, Switzerland. [2]Cryo-EM Knowledge Hub, ETH Zürich, Zürich, Switzerland. [3]Bio-Protection Research Centre, Lincoln University, Lincoln, Christchurch, New Zealand. [4]AgResearch, Resilient Agriculture, Lincoln Research Centre, Christchurch, New Zealand. [5]These authors contributed equally: Charles F. Ericson, Pavel Afanasyev, Yun-Wei Lien. ✉e-mail: pilhofer@biol.ethz.ch

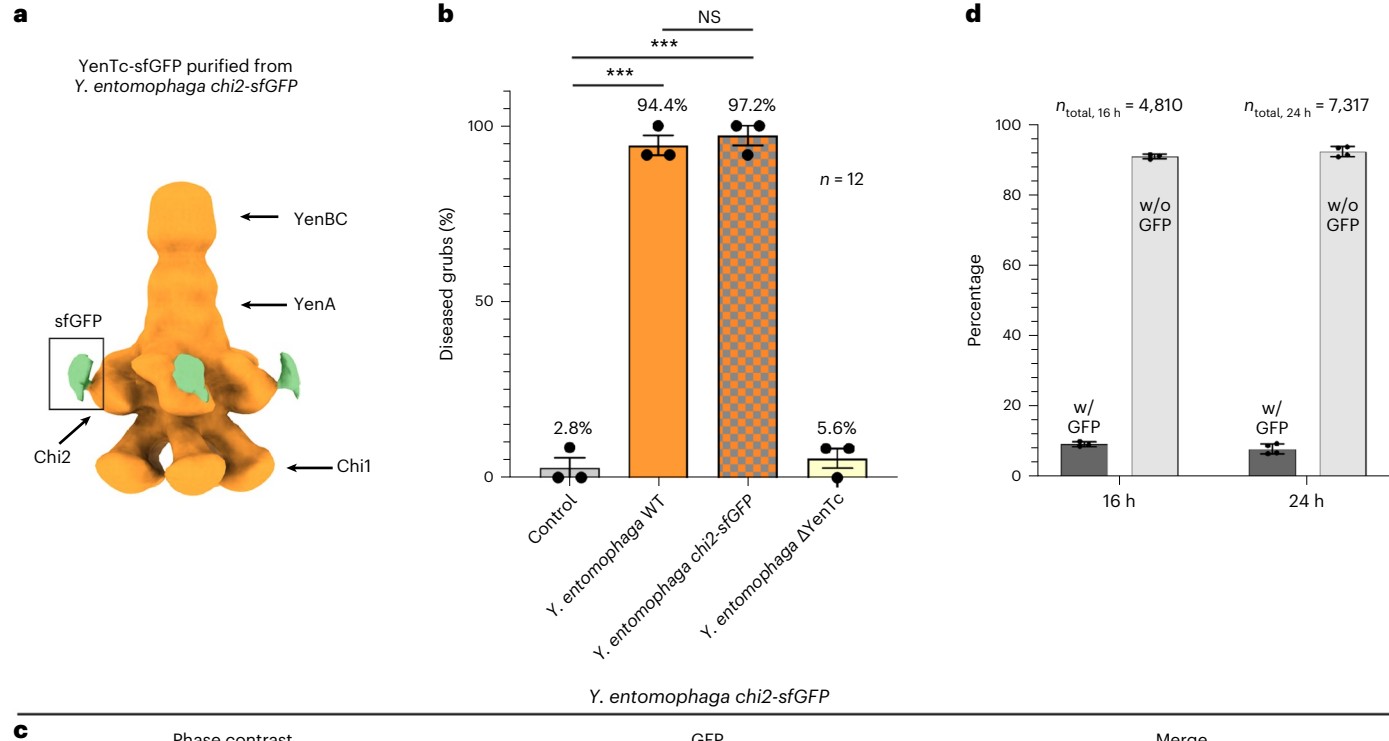

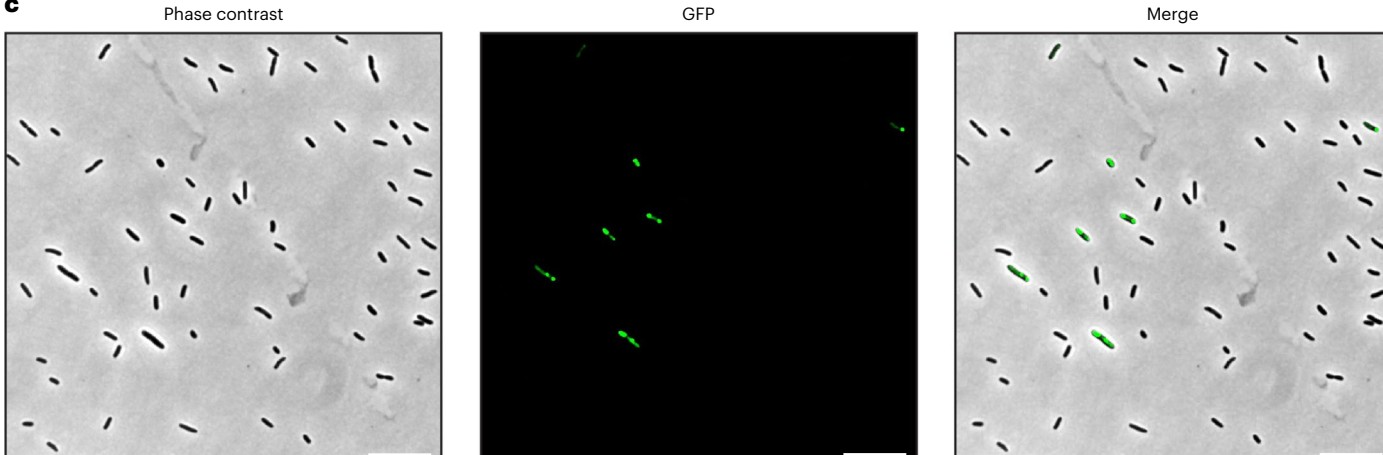

**Fig. 1 | Engineering of a functional *chi2-sfGFP* fusion reveals YenTc-sfGFP expression in a subpopulation. a**, Five-fold rotationally symmetrized subtomogram average of YenTc holotoxin purified from *Y. entomophaga chi2-sfGFP* strain. The positions of the main components of YenTc are indicated. An additional density was observed in the periphery of Chi2 (green/boxed), indicating the presence of flexible sfGFP at this position (see also Extended Data Fig. 1 and Supplementary Fig. 1). **b**, *C. giveni* larvae were fed with carrot food cubes with culture filtrates of different *Y. entomophaga* strains or without filtrate (control). Intoxication assay was conducted 6 days after feeding. The assay shows the pathogenic activity of WT and *chi2-sfGFP* strains. Bar height denotes the mean ratio percentage (±s.e.m.) of diseased grubs derived from biological triplicate experiments (*n* = 12 larvae) for each treatment. Individual percentage for each experiment is shown as dots. Significance was determined through a pairwise comparison using a binomial-logit generalized linear model. Relative to

the untreated control, *Y. entomophaga* WT and *chi2-sfGFP* significantly differed (*P* < 0.001), while ΔYenTc did not differ (*P* = 0.563). No significant difference was noted between *Y. entomophaga* WT and *chi2-sfGFP* (*P* = 0.563). ***\*P* < 0.001. NS, not significant. **c**, fLM of *Y. entomophaga chi2-sfGFP* revealed the expression of Chi2-sfGFP in only a subpopulation of cells. GFP signal was observed as foci at the cell poles or distributed in elongated cells (for more examples, see Extended Data Fig. 2 and Supplementary Fig. 2). Scale bars, 10 μm. **d**, Quantification of fLM imaging revealed that 9.1% and 7.7% of the cells expressed Chi2-sfGFP at 16 and 24 h, respectively. Quantification was based on GFP signal observed in intact cells. *n* indicates the total number of analysed cells. Bar size denotes the mean ratio percentage (±s.d.) of cells with (w/ GFP) and without GFP signal (w/o GFP) derived from biological triplicate (16 h) and quadruplicate (24 h) experiments with $n_{\text{total,16h}}$ = 4,810 and $n_{\text{total,24h}}$ = 7,317 (*n* > 1,000 cells for each experiment). Individual percentage for each experiment is shown as dots.

*Y. entomophaga* is active against a wide range of insect pests[21,23] and produces the ~2.4 MDa Tc toxin 'YenTc'[23] as the main virulence factor. YenTc is referred to as a type II Tc toxin due to the unique architecture of its pore forming 'YenTcA' component. This component is assembled by four proteins, YenA1 and YenA2 (forming the typical TcA complex) and two additional functional endochitinases (Chi1 and Chi2)[13]. The gene cluster is located on a pathogenicity island with

the remaining components TcB (*yenB1*) and two TcC homologues (*yenC1* and *yenC2*) that can both alternatively be incorporated into the holotoxin (Extended Data Fig. 1a). YenC1 has been predicted as a necrotizing factor acting on Rho GTPases, and YenC2 has been assigned to the YwqJ-like deaminase family[9,23]. Recent studies proposed that Chi1 and Chi2 are potential mediators of cell surface recognition and are important for target cell specificity by harbouring lectin activity[13].

## Results

### An sfGFP fusion reveals YenTc expression in a subpopulation

To investigate YenTc expression at a single-cell level, we set out to engineer a fluorescently tagged mutant. Chi2 is an integral part of YenTc[13] (Extended Data Fig. 1a) and we rationalized that placing a peripherally localized tag may not interfere with holotoxin formation. We therefore fused a super-folder green fluorescent protein (sfGFP) tag (~26.8 kDa) with the C terminus of Chi2 (~69.7 kDa) using an 'Ala₃Gly₃' linker. SDS–polyacrylamide gel electrophoresis (PAGE) analysis and western blotting of the *chi2-sfGFP* mutant culture's supernatant and cell pellet confirmed the expression and expected molecular weight (~96.5 kDa) of Chi2-sfGFP (Extended Data Fig. 1b). To test whether the mutant was able to assemble holotoxins, we purified YenTc from WT and YenTc-sfGFP from the *chi2-sfGFP* mutant cultures and determined structures by cryo-electron tomography (cryoET) and subtomogram averaging[24]. Both cultures produced fully assembled holotoxins. The comparison of both structures revealed additional densities adjacent to Chi2 in the *chi2-sfGFP* mutant, indicating the presence of sfGFP, although in a flexible position due to the used linker (Fig. 1a and Extended Data Fig. 1c–k). Mass spectrometry confirmed the presence of all YenTc components as well as sfGFP (Supplementary Fig. 1a). Filtrates derived from cultures from the *chi2-sfGFP* mutant, the WT and a toxin deletion (ΔYenTc, lacking the entire Tc gene cluster) were tested for their insecticidal activity towards *Costelytra giveni* larvae. *chi2-sfGFP* mutants as well as WT culture filtrates were highly toxic towards the larvae, whereas ΔYenTc strain and negative control were innocuous (Fig. 1b and Supplementary Fig. 1b). Since previous studies have shown that effective intoxication is only achieved by fully assembled holotoxins[19], we conclude that the tag does not interfere with holotoxin formation and function.

Next, we set out to investigate Chi2-sfGFP expression at a single-cell level by fluorescence light microscopy (fLM). Based on previous reports[25], YenTc expression is observed in stationary-phase cultures at 25 °C. We therefore recorded images of cells, in random locations, that were grown at 25 °C for 16 h or 24 h. Interestingly, we detected fluorescent signal only in 9.1% ($n_{16h}$ = 4,810) and 7.7% ($n_{24h}$ = 7,317) of the cells (Fig. 1c,d). Fluorescent signals were detected either at cell poles, distributed inside somewhat enlarged cells or distributed in potentially lysing cells (Extended Data Fig. 2 and Supplementary Fig. 2).

### Large numbers of YenTc toxins are assembled in lysing cells

To study YenTc assembly in situ, we set up a workflow that integrates and correlates cryoET and cryo-fluorescence light microscopy (cryoLM). Since plunge-frozen cells were too thick for direct cryoET imaging, we first prepared thin lamellae of bacterial lawns by cryo-focused ion beam (cryoFIB) milling[26]. These lamellae were first imaged by cryoLM, which guided subsequent cryoET data collection.

CryoLM imaging of lamellae with *chi2-sfGFP* cells confirmed our previous observation that only a subpopulation exhibited a fluorescent signal (Fig. 2a,b,d,e). We then selected areas that showed cells with and without fluorescent signal for cryoET data collection. Cryo-tomograms of these regions of interest (ROI) showed either cells with distorted cell envelopes seemingly in the process of cell lysis, or fully intact cells (Fig. 2c,f). Closer inspection of cryo-tomograms of the lysing cells showed distorted/ruptured cell envelopes, no apparent ribosomes and a large number of YenTc-like densities (Fig. 2c,f,g,i). Depending on the orientation, these YenTc-like densities showed the characteristic shape and size of YenTc toxins, including five-fold symmetry in cross-sectional views. In contrast to lysing cells, closer inspection of intact cells showed typical features such as an intact cell envelope, a dense cytoplasm, putative ribosomes and the absence of apparent YenTc-like densities (Fig. 2c,f,g). Interestingly, some intact cells (but never lysed cells) showed elaborate bundles of cytoplasmic filaments with unknown identity (Fig. 2f). To complement the visual approach to identify the location of putative YenTc in an unbiased manner, we applied template matching[27] to a cryo-tomogram with intact and lysed cells to computationally search for densities with structural similarities to YenTc. This confirmed the presence of YenTc almost exclusively in lysing cells (Fig. 2h and Extended Data Fig. 3). Subtomogram averaging of particles that were identified by template matching resulted in a three-dimensional (3D) reconstruction with high similarity to the 3D reconstruction of the purified YenTc-sfGFP fusion (Fig. 2h). Quantification of the correlation of cryoLM with cryoET data showed that lysing cells always exhibited GFP signal. Intact cells with filaments also always showed GFP signal, but the signal did not correlate with the filaments. Intact cells without filaments never showed GFP signal (Fig. 2j).

To rule out that the GFP tag affected assembly and/or release, we imaged cryoFIB-milled WT cells by cryoET. Similar to the GFP-tagged strain, we observed three classes of cells: lysing cells, intact cells with filaments and intact cells without filaments (Fig. 3a–d). Approximately 32% of total cells showed signs of lysis (Fig. 3e; the discrepancy to the 10% observed by LM may be due to differences in sample preparation) and almost all cryo-tomograms of lysing cells contained YenTc-like particles (Fig. 3e). We repeated the template matching from above and detected YenTc hits almost exclusively in lysing cells (Fig. 3f,g and Extended Data Fig. 4). Again, subtomogram averaging of particles that were identified by template matching confirmed a YenTc-like 3D reconstruction (without the additional density for sfGFP, Fig. 3g). To further confirm the identity of these densities as YenTc, we also imaged ΔYenTc cells. Closer inspection of these mutant cells showed that as expected, lysing cells never contained any YenTc-like densities ($n$ = 33, Supplementary Fig. 3).

On the basis of cryoLM correlation, cryoET, mass spectrometry, template matching, subtomogram averaging and mutant analysis, we therefore conclude that YenTc holotoxins are assembled inside cells that are in a state before complete cell lysis, with the assembly taking

**Fig. 2 | YenTc-sfGFP expression correlates with signs of cell death.**
**a,b,d,e,** Representative images of correlated cryoLM and cryoEM projection images of two cryoFIB-thinned lamellae containing *Y. entomophaga chi2-sfGFP* cells. Chi2-sfGFP expression was only observed in a subpopulation of cells. **a,d,** Overview images. **b,e,** Close-up views. Dotted boxes indicate the field of views of the tomograms shown in **c** and **f**. The correlation of the GFP signal to intact cells harbouring cytoplasmic filaments was observed in 4 cases at time point 24 h (total correlated cells $n_{24h}$ = 26) and in 4 cases at time point 16 h (total correlated cells $n_{16h}$ = 45). The correlation to lysing cells at 24 h was observed in 18 cases (total correlated cells $n_{24h}$ = 26) and 34 cases at 16 h (total correlated cells $n_{16h}$ = 45). Scale bars, 1 μm. **c,f,** Representative slices through cryo-tomograms of the ROIs from the corresponding lamellae shown in **b** and **e**. Three classes of cells were observed: Class 1: lysing cells (in **c,f**) that showed cell envelope defects and YenTc-like densities (orange boxes in close-up view). These cells always correlated with GFP signal. Class 2: intact cells that did not show YenTc-like structures (in **c,f**). Most of these cells did not show correlated GFP signal. Class 3: intact cells with filament bundles (in **f**). These cells always showed correlated GFP signal. The positions of the shown close-up views (insets) are indicated by dotted boxes. OM, outer membrane; CM, cytoplasmic membrane. Tomogram slice thickness, 18.04 nm. Scale bars, 100 nm. **g,** A cryo-tomogram of cryoFIB-milled *Y. entomophaga chi2-sfGFP* cells, which was used for subsequent template matching (shown in **h**). The tomogram reveals two intact cells (top and centre) and one lysing cell (bottom). Tomogram slice thickness, 18.04 nm. Scale bar, 100 nm. **h,** The cryo-tomogram from **g** was used for template matching to search for YenTc-like densities. Shown is a 3D segmentation of the cryo-tomogram. YenTc-like densities (orange) were almost exclusively detected in the lysing cell (green). The template matching coordinates were used for subtomogram averaging and resulted in a structure resembling YenTc-sfGFP (inset). See also Extended Data Fig. 3. **i,** Magnified views of the lysing cell shown in **g** (positions indicated by dotted boxes) reveal YenTc-like densities. **j,** Quantification of the cell types seen by cryoET and their correlation with GFP signal in cryoLM at time point 24 h ($n_{w/GFP}$ = 26 cells, $n_{w/oGFP}$ = 36 cells). Lysing cells (69.2%, green) and intact cells with filaments (15.4%, blue with black dots) always showed GFP signal.

place before their release. Interestingly, while these cells appear to be packed with YenTc, they exhibit a low-density region between toxins and the lysing cell envelope, as well as frequent structures spanning the periplasm between cytoplasmic and outer membranes (Extended Data Fig. 5).

**A phage-like lysis cassette mediates YenTc toxin release**

Because YenTc was only detected in cells with signs of cell lysis, we attempted to determine the fate of Chi2-sfGFP-expressing cells by timelapse imaging. We frequently observed cells with sfGFP signal at the poles. A typical pattern over time was that sfGFP-positive cells

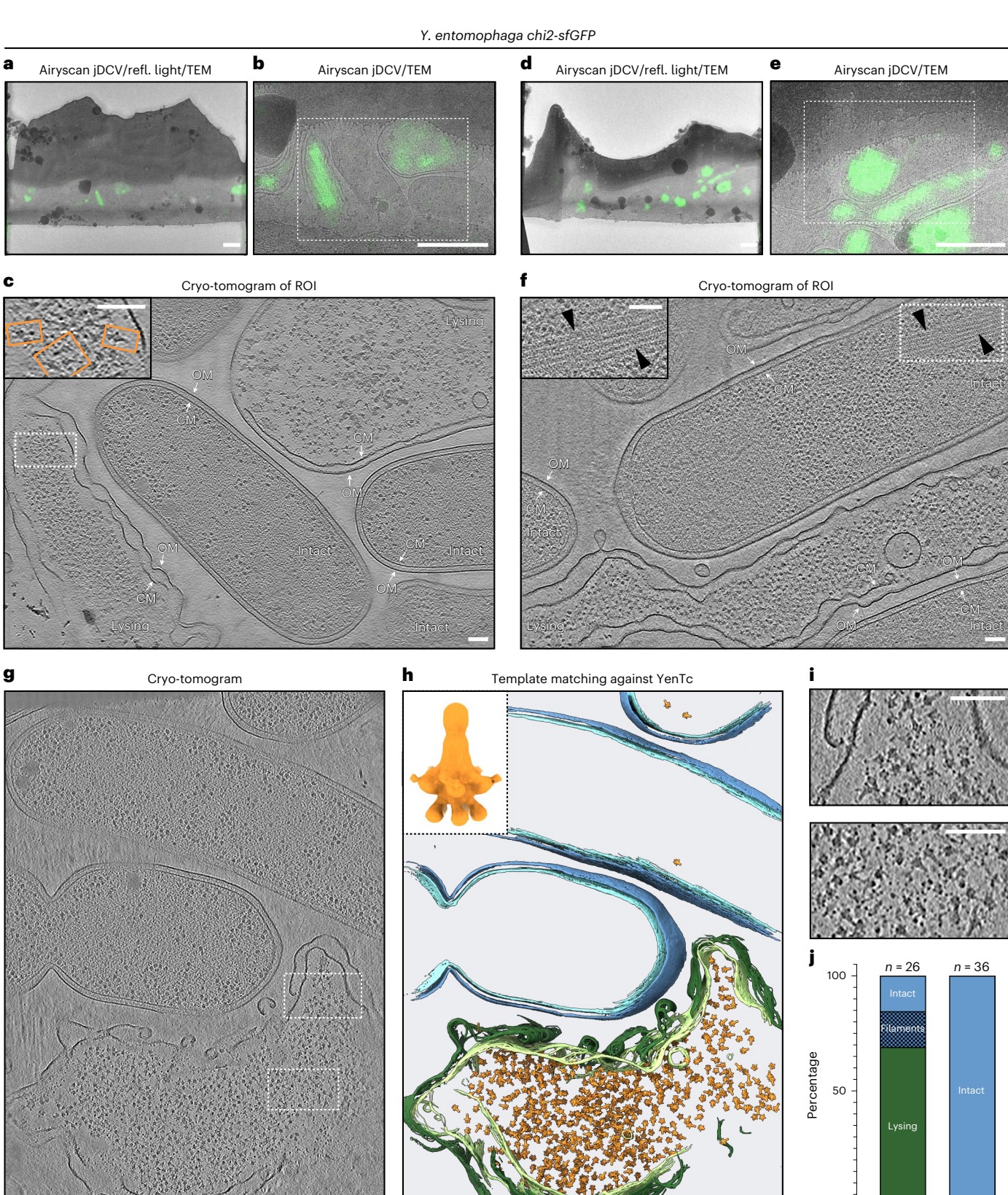

*Y. entomophaga WT*

**Fig. 3 | Lysing *Y. entomophaga* WT cells contain large numbers of assembled YenTc particles. a**–**d**, Representative slices through cryo-tomograms of cryoFIB-milled *Y. entomophaga* WT cells. Again, three classes of cells were observed: Class 1: lysing cells (in **a**,**c**) that showed cell envelope defects and YenTc-like densities (orange boxes). Class 2: intact cells that did not show YenTc-like structures. Class 3: intact cells with filament bundles (in **b**,**d**, indicated by arrowheads). **c**,**d**, Close-up views of dotted boxes in **a** and **b**. Tomogram slice thickness, 18.04 nm. Scale bars, 100 nm. **e**, Quantification of the cell types seen by cryoET (left) and quantification of the abundance (95.2%) of YenTc-like particles in lysing cells (right) ($n_{total}$ = 132 cells, $n_{lysing}$ = 42 cells). **f**, The cryo-tomogram shown in **a** was used for template matching to search for YenTc-like densities. Shown are all curated hits (orange) and a representative tomographic slice (see also Extended Data Fig. 4). Tomogram slice thickness, 18.04 nm. Scale bar, 100 nm. **g**, A 3D segmentation of the cryo-tomogram shown in **a** and **f** displaying YenTc in orange, lysed cell envelopes in green and intact cell envelopes in blue. Note that YenTc hits were almost exclusively detected in the lysing cells. Inset: subtomogram average of subvolumes extracted from template matching coordinates. As expected, densities for GFP are absent.

elongated and increased in diameter, and then showed whole-cell fluorescence. This was often followed by a loss of phase contrast and leaking of sfGFP signal from the cell, which indicated cell lysis (Fig. 4a). Next, we quantified the correlation between cell lysis and sfGFP signal in individual fLM images (Extended Data Fig. 2 and Supplementary Fig. 2). In cultures that were grown for 16 h or 24 h, lysed cells showed the presence of sfGFP signal in 72.6% and 92.3% of the cases ($n_{16h}$ = 77, $n_{24h}$ = 193), respectively.

We then set out to determine the mechanism of cell lysis. A previous bioinformatic approach detected a phage-like lysis cassette (LC) that was often found in the vicinity of Tc gene clusters in different bacterial strains and postulated to be involved in general protein secretion[28]. This is consistent with a report from *Y. enterocolitica*, where a phage-related holin/endolysin pair was suggested to be important for the release of its insecticidal toxin complex[29]. A similar LC has been identified by a transposon screen in *Y. entomophaga* WT, but the LC was not encoded in the vicinity of the YenTc gene cluster[25,30]. This LC encodes a holin (*holA*), an endolysin (*pepB*) and two spanin (*rz*, *rz1*) genes (Fig. 4b), and was shown to be required for general exoprotein release[30].

To assess the role of the lysis cassette in YenTc release, we investigated a mutant[30] lacking *holA*, *pepB*, *rz* and partial *rz1* (ΔLC). Cell lysate and supernatant of the ΔLC mutant were probed by western blotting using an anti-YenA1 antibody (Fig. 4c). While cytoplasmic expression of YenA1 was verified, YenA1 was not detected in the exoproteome. YenTc release is therefore mediated by the LC.

**Disintegration of cell envelope triggers holotoxin assembly**

Interestingly, light microscopy imaging of the ΔLC mutant showed a substantial elongation of the cells in comparison with the WT, with some cells occasionally reaching a length of up to 80 μm (Fig. 4d and Extended Data Fig. 6). We decided to further characterize ΔLC mutant cells by cryoET imaging. Consistent with the hypothesis that the lysis cassette mediates cell lysis, the ΔLC mutant showed primarily intact cells and only rarely showed signs of lysis. 81.9% of the cells at 24 h ($n_{24h}$ = 116 cells) and 82.9% of the cells at 16 h ($n_{16h}$ = 82 cells) contained the filament bundles that were previously observed in WT and the *chi2-sfGFP* mutant (Fig. 4e). Surprisingly, despite the detection of YenTc components by western blotting, intact cells were devoid of any apparent YenTc-like densities.

To test whether externally induced cell lysis would result in the assembly of holotoxins, we first pelleted ΔLC cells, lysed them enzymatically by the addition of a lysozyme-containing buffer and then pelleted YenTc toxins from the supernatant. Imaging such a preparation revealed massive amounts of assembled holotoxins (Fig. 4f). Second, we were interested in whether a faster process of breaking the cells would show similar results. We therefore froze a pellet of ΔLC mutant cells in liquid nitrogen and subjected the frozen cells to mechanical shearing by steel balls at cryogenic temperatures using a 'mechanical cryo-mill'. Throughout the process, the sample was kept at cryogenic temperature. The resulting frozen and processed powder was then resuspended in Tris buffer and immediately subjected to plunge freezing for subsequent cryoET imaging. Remarkably, this preparation showed the absence of any filaments and the presence of assembled holotoxins inside sheared cells and in the extracellular space (Fig. 4g). We therefore conclude that YenTc components are expressed in the ΔLC mutant; however, holotoxin assembly is triggered upon inflicting defects in the cell envelope, accompanied by rapid disassembly of the filaments.

**Cells are primed with virulence factors before cell lysis**

The above characterization indicated that the ΔLC mutation may arrest the cells in an intermediate state, in which YenTc subunits are expressed but not yet assembled into the holotoxin, concomitant with the expression of elaborate filament bundles. This cellular state could represent (1) the class of intact cells with filaments that were seen in WT and (2) the class of intact cells with filaments and GFP signal that were seen in the *chi2-sfGFP* strain. We therefore set out to characterize this interesting cellular state in more detail.

First, we focused on the identity of the filament bundles. Besides the observed differences in their abundance, their filament ultrastructures showed no notable differences in WT, *chi2-sfGFP* and ΔLC strains. The individual filaments had a diameter of ~17 nm and were seen in bundles of up to ~18 parallel filaments. To test whether the filaments were composed of one or several components of the YenTc gene cluster, we generated a deletion mutant lacking the entire YenTc toxin gene cluster (*chi1/yenA1/A2/chi2/yenB/yenC1/C2*) in the background of ΔLC (because the filaments were most abundant in this background) (Extended Data Fig. 7a). Imaging these mutant cells exhibited the presence of filament bundles (Extended Data Fig. 7b) in 75% of the cells (*n* = 100), ruling out that the filaments comprised YenTc components.

To understand whether the coexistence of YenTc and filaments were both in response to a similar gene expression programme, we tested their expression in WT and ΔLC cultures grown at 25 °C and 37 °C, respectively. Consistent with previous data[23,25], YenA1 was not detectable in cells grown at 37 °C (Extended Data Fig. 8a,b). Interestingly,

**Fig. 4 | YenTc release is mediated by a phage-like LC. a**, Timelapse fLM of *Y. entomophaga chi2-sfGFP* cells showing that sfGFP-positive cells often elongate and lyse during the course of imaging. Black, orange and white arrowheads point to three different lysis events. Scale bars, 10 μm. **b**, Schematic of the LC gene cluster in *Y. entomophaga*, which comprises a holin, an endolysin and two overlapping spanin open reading frames (gene names are indicated). This gene cluster has been previously identified, being involved in general exoproteome release[30]. **c**, Western blot analysis against YenA1 and RecA (loading control) of *Y. entomophaga* WT and ΔLC mutants showed that both strains expressed YenA1, but release into the supernatant (sup.) was abolished in the ΔLC mutant. The experiment was repeated five times with similar results. **d**, fLM of *Y. entomophaga* ΔLC showed a distinct elongated cell shape. Scale bar, 10 μm. The experiment was repeated five times with similar results. **e**, Slice through a cryo-tomogram of cryoFIB-milled *Y. entomophaga* ΔLC cells. Cells were elongated and did not reveal apparent YenTc-like densities. Of the cells, 81.9% ($n_{24h}$ = 116 cells) showed elaborate filament bundles (arrowheads) similar to those seen in *Y. entomophaga* WT and *chi2-sfGFP*. Tomogram slice thickness, 18.04 nm. Scale bar, 100 nm. **f**,**g**, *Y. entomophaga* ΔLC mutant cells were subjected to external cell lysis by lysozyme treatment (**f**) or a mechanical cryo-mill (**g**). Both procedures resulted in the absence of any filaments and the presence of YenTc particles (orange boxes). Shown are cryo-tomograms of a crude lysate preparation (**f**) and *Y. entomophaga* ΔLC mutant cells that were subjected to mechanical cryo-milling (**g**). Note the YenTc particles inside and outside cells with disrupted cell membranes. Tomogram slice thickness, 18.04 nm. Scale bar, 100 nm.

cryoET imaging showed that filaments were also entirely absent from both strains ($n_{\Delta LC}$ = 134, $n_{WT}$ = 175) grown at 37 °C (Extended Data Fig. 8c,d). These insights allowed us to design a proteomics experiment to compare the proteomes of cultures grown at 25 °C and 37 °C. We performed parallel label-free proteomics quantification with

subsequent differential expression analysis (DEA) with WT and ΔLC cell lysate and used 25 °C as the control standard (ΔLC cells show a much higher abundance of filament bundles). As expected, the comparison of ΔLC datasets revealed YenTc components among the top hits, being highly abundant at 25 °C and downregulated or not detectable at 37 °C

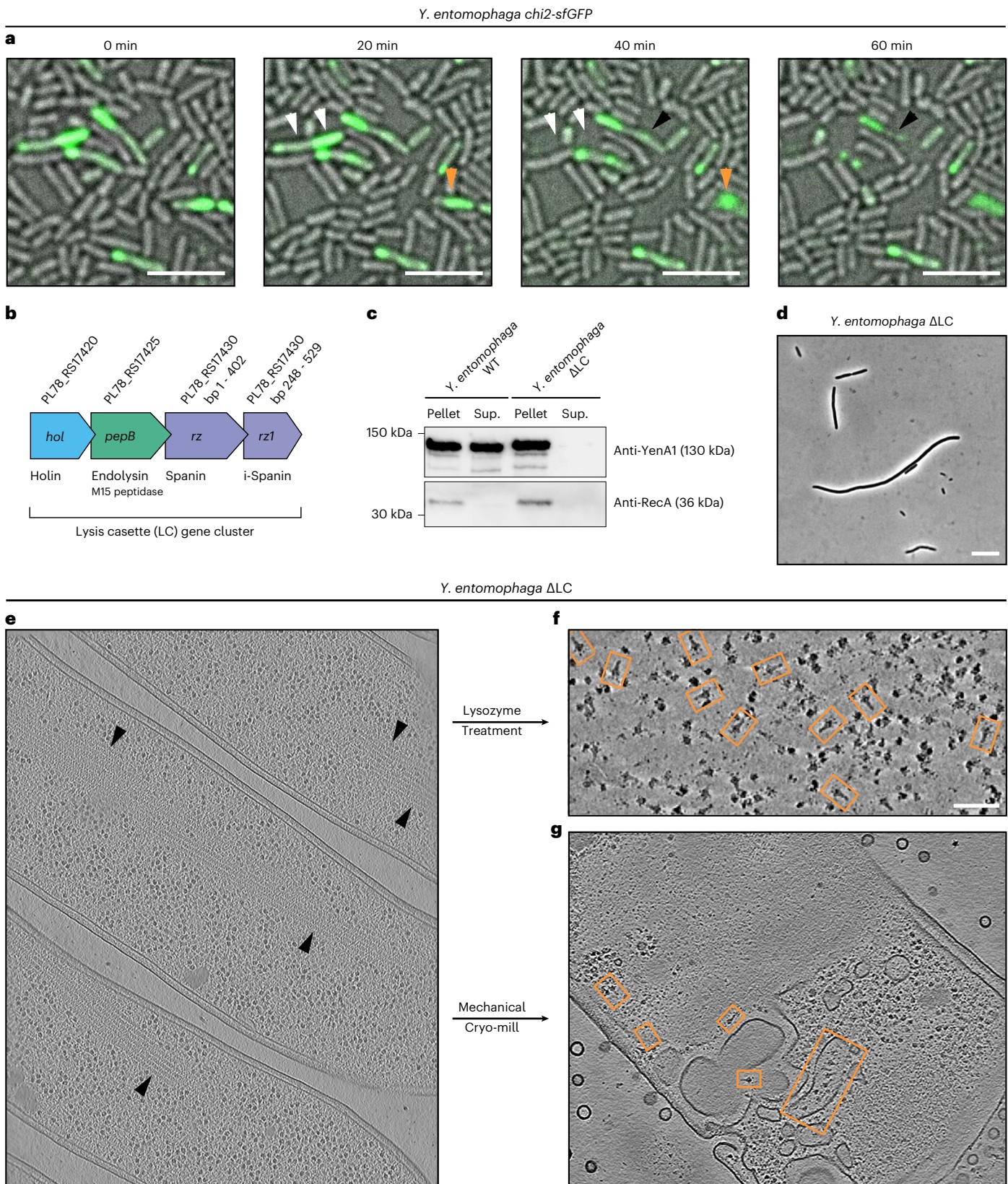

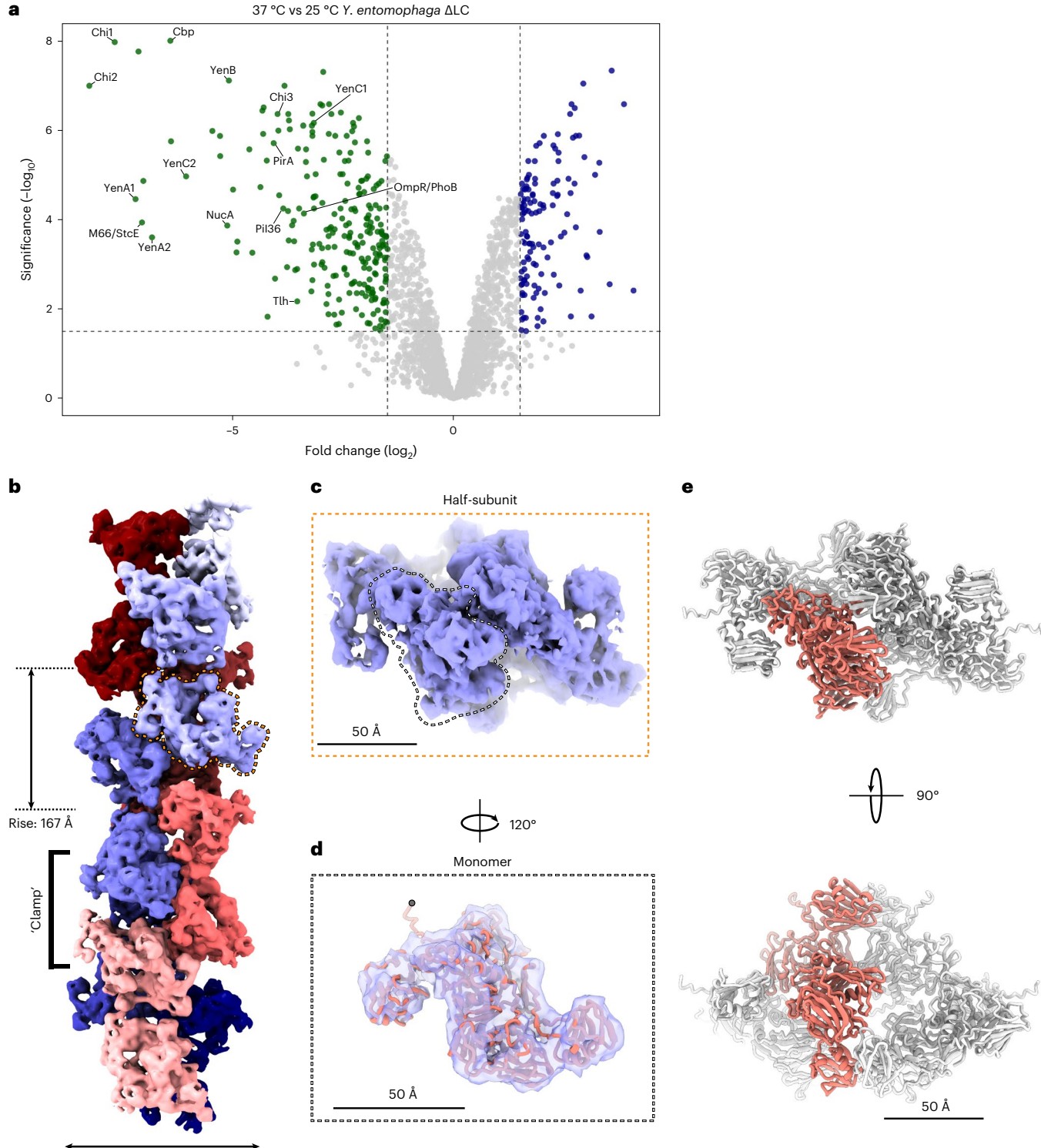

**Fig. 5 | Primed cells express multiple virulence factors and filaments composed of the protease M66/StcE. a**, *Y. entomophaga* ΔLC cells were grown at 37 °C or 25 °C (control standard), analysed by mass spectrometry and DEA. The volcano plot shows the −log₁₀ FDR as a function of log₂ FC when comparing the ΔLC samples grown at 37 °C with those grown at 25 °C. All negative FC values (log₂ base), their FDR values as well as their confidence intervals can be found in Supplementary Table 1. Note that proteins that were significantly more abundant at 25 °C are shown in the negative FC. **b**, Subtomogram average of a filament segment from *Y. entomophaga* ΔLC cells. The filament consists of two identical intertwined strands (indicated by red and blue). The consecutive subunits in these strands along a helical filament are shown in different shades of red and blue. The neighbouring subunits form clamp-like arrangements

(clamp region indicated). The helical parameters (rise 167 Å, twist 88.9°) were imposed to generate the shown 3D reconstruction for illustration purposes. One half-subunit is outlined by a dashed orange line. **c,d**, The map of the half-subunit was searched for fits with candidate structures and retrieved best hits for four M66/StcE monomers. Shown is the half-subunit/tetramer (**c**, along the two-fold axis) and the monomer (**d**) of the final cryoEM map (see also Extended Data Fig. 10). The rigid-body fitting of the M66/StcE monomer (indicated by white dashed line in **c**) into the map is shown in **d**. Residues 1–24 were not detected by mass spectrometry and are therefore not shown; grey disc indicates N-terminal methionine at position 25. **e**, Atomic model resulting from the above analyses, showing the M66/StcE homotetramer (same orientation as in **c**). One of the monomers is represented in salmon. **c** and **e** are at the same scale.

(Fig. 5a and Supplementary Table 1). Beyond all YenTc components, a set of other proteins showed a similar fold change (FC) ($\log_2$ FC < −3 and false discovery rate (FDR) < 1%; Supplementary Table 1) when compared with the cell culture grown at 25 °C. These included potential virulence factors, such as Chi3 (chitinase)[31], Cbp (chitin-binding protein)[31], PirA (insect-related toxin)[32], Pil36 (pilin), NucA (nuclease)[33], Tlh (hemolysin)[34], M66/StcE (metalloprotease)[35–39] and a regulator of the lysis cassette (OmpR/PhoB-containing protein)[30]. WT cells showed the same trend of the aforementioned proteomic change ($\log_2$ FC < −3, FDR < 1%) upon temperature decrease, but in addition also revealed the upregulation of the endolysin (A0A3S6F4L4) from the LC ($\log_2$ < −2 and FDR < 10%; Supplementary Table 2 and Extended Data Fig. 9).

### The protease M66/StcE forms filaments in primed cells

We then used an integrative approach to determine whether any of the candidates retrieved by mass spectrometry may polymerize into the observed filaments. We therefore picked and computationally analysed 35,332 subvolumes from filaments from 42 cryo-tomograms of ΔLC mutant cells using subtomogram averaging (Extended Data Fig. 10). The resulting 3D reconstruction at ~10 Å resolution (without application of any symmetry) revealed that the filaments are composed of two identical intertwining strands whose subunits are related by a helical symmetry (rise 167 Å, twist 88.9°) (Fig. 5b). The subunits of the neighbouring strands form clamp-like structures, which might contribute to the flexibility of the filaments observed by 3D classification.

The obtained resolution allowed for the segmentation and extraction of half of the subunit (Fig. 5c) for the determination of protein identity. We used AlphaFold2-predicted structures of the candidates that were retrieved by our proteomics analyses and applied rigid-body fitting into the final cryoEM map. The majority of the candidates did not show reasonable fits, revealing either many densities unfilled and/or exposing many residues outside of the map. Notably, two pairs of the candidate M66 (A0A3S6EYX4) appeared to fit very well into the half-subunit, leaving no cryoEM densities unaccounted for (Fig. 5d). The quality of the fit suggested that the filaments are composed of M66 proteins. The rigid-body search tool Colores[40,41] confirmed the four positions for M66 in the half-subunit, which appears as a homotetramer (Fig. 5e).

To validate the identity of the filaments, we confirmed the absence of similar sequences (using BLAST[42]) and structures (using DALI[43,44] and Foldseek[45]) from the WT proteome. Furthermore, we generated a ΔLCΔM66 double mutant (ΔLC background used due to the high abundance of filaments). Imaging these mutant cells (*n* = 103) revealed the absence of M66/StcE filament bundles, while the purification of YenTc from ΔLCΔM66 double mutant cells detected assembled holotoxin particles (Supplementary Fig. 4).

Altogether, we show that filaments are composed of the protein A0A3S6EYX4, having similarities to metalloprotease M66. Interestingly, the homologue StcE represents a secreted virulence factor in enterohemorrhagic *Escherichia coli* (EHEC)[35–38,46].

## Discussion

We identified the phage-like lysis cassette as a key factor for YenTc release. This finding is consistent with previous studies suggesting a role of such a cassette in exoproteome release[22,25] and in the pathogenicity of *Y. enterocolitica*[29]. Furthermore, lysis cassettes are also conserved in other Tc toxin-positive species, including *S. entomophila* and *Photorhabdus* species[28,30]. The association of lysis cassettes with Tc toxin gene clusters points to a conserved mechanism of Tc toxin release.

The release of large macromolecular complexes that cannot be secreted by classical secretion systems is a challenge for bacterial cells and does not only apply to Tc toxins. Accordingly, phage lysis cassettes may also be used for the release of other virulence factors, a mechanism for which the term 'type 10 secretion system' has been proposed[28,47]. Also, protein secretion via membrane vesicle formation depends on such lysis cassettes[48–50]. Another example of a supramolecular

complex that is released in an assembled state are extracellular contractile injection systems (eCIS), which use a phage tail-like apparatus to inject effectors into targets. Based on their close evolutionary relationship to contractile phages, eCIS are thought to be released by the induction of phage-related lysis cassettes[51]. Furthermore, metamorphosis-associated eCIS are also only released by a bacterial subpopulation[52]. Interestingly, Tc toxins and eCIS frequently co-exist in different organisms, including species of *Serratia* and *Photorhabdus*. This may indicate that strains that harbour a lysis cassette may be able to evolve different large complexes that act in the extracellular space.

YenTc toxins are released by a mechanism that leads to killing of a bacterial subpopulation by cell lysis. This fatal outcome for the subpopulation probably represented a strong evolutionary pressure to become as efficient as possible. Our data indicate that this may be achieved by a sequential release mechanism that results in the assembly of YenTc holotoxins and the priming of cells with virulence factors before their release by cell lysis (see model in Fig. 6). At 25 °C, the expression of Chi2 (seen at a single-cell level by GFP tagging and cryoLM imaging, Figs. 1 and 2) and probably other YenTc components (seen at cell-culture level by proteomics, Fig. 5a and Extended Data Fig. 9) is significantly upregulated in a small subpopulation of cells, but not yet assembled into a holotoxin (seen by correlative imaging of *chi2-sfGFP* and ΔLC cells, Fig. 4). This YenTc-positive subpopulation also induces the co-expression of M66 filaments (seen in Fig. 2d–f; discussion below). Furthermore, our mass spectrometry data revealed the upregulation of multiple other factors that are known to play roles in the pathogenicity of different bacterial pathogens (Fig. 5a and Extended Data Fig. 9), including the JHE-like *Photorhabdus* insect-related toxin PirA[32], endonuclease NucA[33], hemolysin Tlh[34], adhesin Pil36 and chitin-binding and/or chitin-degrading enzymes Chi3/Cbp[31]. This notable switch in gene expression primes a subpopulation of cells for the assembly and release of these factors. The increased volume (achieved by elongation/larger diameter) in primed cells (Fig. 4) could represent an adaptation that maximizes the toxic 'load' that can be produced per primed cell.

The next step is induced by the action of the phage lysis cassette. These gene clusters typically mediate bacterial cell lysis by (1) the action of a holin, facilitating access to the periplasm for the endolysin and (2) the action of the endolysin, degrading the peptidoglycan cell wall[53]. Importantly, this process does not lead to the immediate and complete loss of cellular integrity. Instead, we observed 'ghost' cells whose cell envelope has already been disturbed, yet the envelope is intact enough to serve as an 'assembly compartment' for holotoxins. Such ghosts were typically packed with large numbers of assembled YenTc holotoxins (Figs. 2h and 3g, and Extended Data Fig. 5).

Interestingly, similar to the LC in *Y. entomophaga*, some phage lysis cassettes also encode spanin genes[53,54]. These spanins are thought to extend across the periplasmic space. For some spanins, deletion mutants were proposed to result in spherical cells that do not proceed to release phage particles[53,55]. While the structure and mechanism of spanins are unknown, our cryo-tomograms of ghosts frequently revealed densities spanning between cytoplasmic and outer membranes, in particular at constriction sites (see arrows in Extended Data Fig. 5). It is tempting to speculate that these structures represent spanins, which may act as a checkpoint before catalysing the final step in ghost cell lysis. Future analyses of spanin deletion strains could dissect the final step of the sequential release mechanism and confirm the identity of putative spanin densities.

This stepwise mechanism of toxin assembly and release is probably the basis for minimizing the fraction of the total cell population that needs to be killed, since it probably (1) maximizes the number of YenTc toxins and other virulence factors that are produced per cell and (2) maximizes the efficiency of holotoxin assembly inside 'assembly compartments' rather than in the medium. The programmed cell death of a subpopulation may confer a fitness advantage to the remaining bacterial community[56].

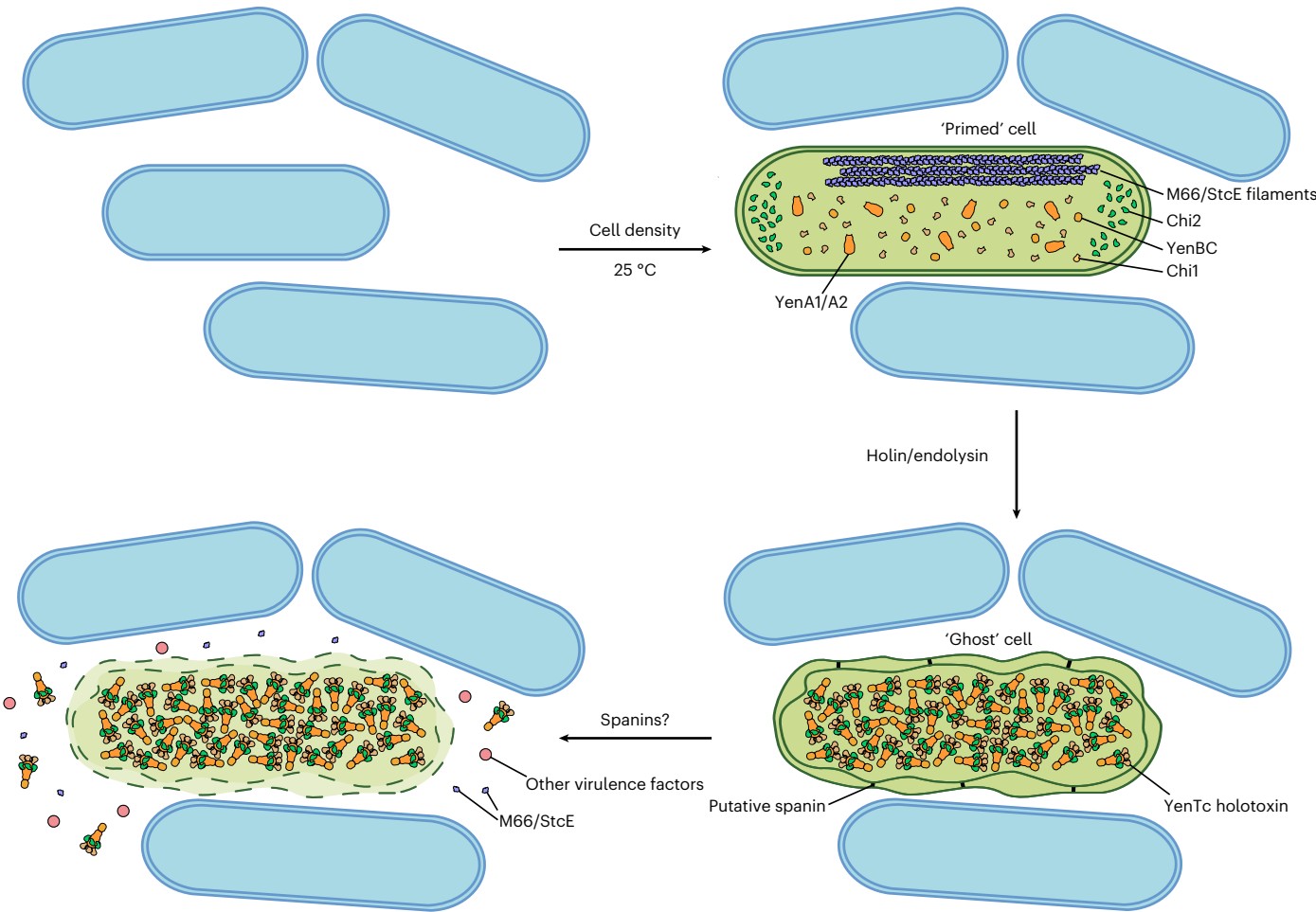

**Fig. 6 | YenTc and other virulence factors are assembled and released by an orchestrated and stepwise mechanism.** Schematic of our hypothetical model. Environmental factors such as cell density and temperature induce a specialized gene expression programme in a subpopulation of cells, resulting in 'primed' cells (green cells) that overexpress individual YenTc components (orange and green) and other potential virulence factors such as M66/StcE filaments (purple).

The action of a phage-like lysis cassette (holin/endolysin) inflicts cell envelope defects, resulting in 'ghost' cells and the disassembly of M66/StcE filaments. Ghosts may serve as compartments for YenTc holotoxin assembly. We speculate that spanins (black) may catalyse the final step in the release of YenTc, M66/StcE and other virulence factors (red).

A remarkable discovery was the presence of filaments in primed cells of the WT, the *chi2-sfGFP* and the ΔLC strains. Our integrated approach allowed us to identify the filaments as being composed of a protein (A0A3S6EYX4), which has similarities to M66 metalloproteases and has not been characterized in *Yersinia*. Polymerization of M66 may keep the enzyme in an inactive and non-toxic state for the producer, analogous to metabolic enzymes that are inactivated by polymerization[57].

One possible hypothesis for the function of M66 is a role in YenTc assembly. We exclusively observed either M66 filaments 'or' holotoxins, but never both together in the same cell. At the same time, both proteins were clearly co-expressed inside the same cell (see Figs. 2d–f and 4e–g). Furthermore, inflicting external cell envelope stress (by lysozyme or mechanical cryo-mill) to ΔLC cells (containing highly abundant M66 filaments; Fig. 4e) led to the disassembly of M66 filaments and the concomitant assembly of YenTc holotoxins (Fig. 4f,g). The observation of assembled holotoxins in a purification from ΔLCΔM66 cells (Supplementary Fig. 4) indicates that such proteolytic cleavage may not be required for the YenTc assembly process. Future studies will test whether the M66 proteolytic activity may be required for YenTc efficiency or processing of other virulence factors in primed cells.

Alternative to the processing of bacterial proteins, the target of M66 could be components of the host. In fact, M66/StcE homologues function as a virulence factor in EHEC[35–39] and are also found in *Vibrio*[58], *Enterovibrio*, *Aliivibrio*, *Pseudomonas*, *Pectobacterium*, *Shewanella* and *Aeromonas* species. In EHEC, StcE is secreted via the Type II Secretion System and supports penetration into the host and adherence to epithelial gut cells[36,37,39,46] by proteolytic remodelling of the mucosal lining[35,38]. TagA, another homologue in *V. cholerae*, is a mucinase, which modifies host cell surface molecules during infection[58]. In *Y. entomophaga*, M66 might play a similar role in insect pathogenicity by mediating bacterial adhesion to host cells or making host cells accessible for the binding of YenTc or other virulence factors.

## Methods

### Bacterial strains

*Y. entomophaga* MH96 (WT) and derivatives were cultured in liquid Luria-Bertani (LB) medium at 25 °C with shaking at 200 r.p.m. or grown for 24 h unless indicated otherwise, or grown on LB medium solidified with 1.5% (w/v) Difco agar. Indicated mutant strains were cultured in LB media supplemented with antibiotics at the following concentrations: 100 μg ml⁻¹ kanamycin and 25 μg ml⁻¹ chloramphenicol.

## Generation of *Y. entomophaga* mutants

To generate the derivatives of *Y. entomophaga* MH96 and ΔLC, 1,000-bp-long fragments of the upstream and downstream regions of the gene were amplified and cloned into the suicide vector pDM4 (ref. 59) or pCVD443 and transformed into *Y. entomophaga* WT or ΔLC by electroporation (for primers, see Supplementary Table 3). The positive colonies were then streaked onto fresh LB plates containing kanamycin for the pCVD443 integration, or kanamycin and chloramphenicol for pDM4. The cultures were streaked out several additional times to ensure the purity of the prospective mutant genotype. Colony PCR was used to confirm the insertion of the plasmid. To select the colonies with double crossing over, a single colony with the plasmid inserted in the genome was cultured in LB without antibiotics (for WT) and kanamycin (for ΔLC) for two nights and spread onto LB agar plates with kanamycin and 15% sucrose. The plates were incubated at 30 °C until colonies formed. Loss of the plasmid was confirmed by selecting the bacteria that did not grow on the LB plates with kanamycin and chloramphenicol but grew on the plates with only kanamycin. Mutations were confirmed by PCR of the genomic DNA. The mutants were further confirmed by western blotting and mass spectrometry (MS) analysis.

## Purification of YenTc toxin particles

YenTc toxin particles from *Y. entomophaga* cultures were purified as previously published[13]. Volumes of 50 ml of an overnight culture of *Y. entomophaga* WT and derivatives were pelleted. Supernatant was collected and filtered two times through a 0.2 μm filter. The supernatant was then subjected to ammonium sulfate precipitation to a final concentration of ~70% w/v ammonium sulfate. The resulting precipitate was resuspended in Tris buffer (25 mM Tris, 150 mM NaCl, protease inhibitor cocktail (Roche), pH 7.5). Another method of purifying YenTc particles was performed using lysis of the cell pellet. Here, cell pellets from the same cell culture were lysed for 1 h at 37 °C, shaking in the lysis buffer (50 mM Tris-HCl, 150 mM NaCl, 0.5x CellLytic B (Sigma-Aldrich), 1% Triton X-100, 200 μg ml$^{-1}$ lysozyme, 50 μg ml$^{-1}$ DNAse I, protease inhibitor cocktail (Roche), 5 mM MgCl$_2$, pH 7.5). Cell debris was removed by centrifugation (15,000$g$, 15 min, 4 °C) and cleared lysates were subjected to ultra-centrifugation (15,0000$g$, 1 h, 4 °C) with a 2 ml 40% sucrose cushion. Pellets were resuspended in 100 μl resuspension buffer (25 mM Tris, 150 mM NaCl, protease inhibitor cocktail (Roche), pH 7.5). Proteins in the toxin particle preparations were identified by mass spectrometry at the Functional Genomics Center Zürich (FGCZ).

## SDS–PAGE and western blotting

Bacterial cultures were incubated for 16 or 24 h at 25 or 37 °C. Optical density (OD)$_{600}$ was adjusted to 1.5 and 2 ml of the bacterial culture was pelleted by centrifugation at 3,000$g$ for 10 min. Supernatants of the cultures were sampled and the cell pellet was further resuspended in 100 μl of lysis buffer (20 mM Tris, 150 mM NaCl, pH 7.4; 5 mM MgCl$_2$, 200 μg ml$^{-1}$ lysozyme; 50 μg ml$^{-1}$ DNAse I, 0.5x CellLytic B (Sigma-Aldrich); 1% Triton X-100, protease inhibitor cocktail (Roche)). After 1 h incubation at 37 °C, the cell debris was removed by centrifugation at 16,000$g$ for 10 min at 4 °C. The supernatant of this lysate was used as the cell pellet (lysate) sample. The samples were denatured for 5 min at 95 °C in 1× Laemmli sample buffer (Bio-Rad) before loading on a 4–20% gradient precast protein gel (Bio-Rad). Gel electrophoresis was carried out at a constant voltage of 200 V in SDS running buffer (Tris/glycine/SDS, Bio-Rad) for 35–40 min.

The gels used for western blotting were transferred onto nitrocellulose membranes. Membranes were blocked with 5% milk in TBS-T (50 mM Tris-HCl pH 7.6, 150 mM NaCl, 0.1% Tween-20) for 1 h at room temperature. Membranes were incubated with 1 μg ml$^{-1}$ rabbit polyclonal antibody against YenA1 (antigen name: YenA1 940-1164, antigen sequence: MGVERSVVPLQLRWLGSNVYSVLNQVLNNTPTDISSIVPKLS-ELTYSLLIYTQLINSKLNKEFIFLRLTQPNWLGLTQPKLSTQLSLPEIYLIT-CYQDWVVNANKNEDSIHEYLEFANIKKTEAEKTLVDNSEKCAELLAEILAW-

DAGEILKAASLLGLNPPQATNVFEIDWIRRLQTLSEKTMISTEYLWQMG-DLTENSEFSLKEGVGEAVMAALKAQGDSDNVHHHHHH) (GenScript), 1:7,500 rabbit polyclonal antibody against GFP (ab183734, abcam) or 1:1,000 rabbit polyclonal antibody against RecA (ab63797, Abcam) and with 1:5,000 secondary HRP-conjugated goat anti-rabbit IgG (31460, Invitrogen) in 1% milk in TBS-T for 1 h. Between and after antibody treatments, membranes were washed three times for 15 min to prevent unspecific binding. Signals were detected using a chemiluminescent substrate (1705061, ECL, Bio-Rad or RPN2105, ECL).

## Mass spectrometry analyses

Purified YenTc of *Y. entomophaga* WT and *chi2-sfGFP* were sent in solution to the FGCZ, which performed the mass spectrometry and the subsequent analysis of primary data. Proteomics samples were prepared by trichloroacetic acid precipitation, followed by trypsin digestion. For trichloroacetic acid precipitation, proteins were precipitated with trichloroacetic acid (Sigma-Aldrich) at a final concentration of 5% and washed with ice-cold acetone. Samples were then air dried and dissolved in 10 mM Tris and 2 mM CaCl$_2$, pH 8.2. Samples were then enzymatically digested using trypsin. These digested samples were dried and dissolved in 20 μl double distilled water with 0.1% formic acid. Samples were transferred to autosampler vials for liquid chromatography–tandem mass spectrometry analysis (LC–MS/MS). For each sample, 3 μl were injected on a nanoAcquity UPLC coupled to a Q-Exactive mass spectrometer (Thermo Fisher). The acquired MS data were converted to a Mascot Generic File format (.mgs files). Identification of proteins was performed using the Mascot search engine (Matrixscience). The spectra were searched against the *Y. entomophaga* protein database.

All protein identification results were visualized using the Proteome software 'Scaffold'.

## Sample preparation for label-free proteomics quantification

Lysate samples were prepared as described above. For each sample (lysate of WT or LC cells grown at 25 or 37 °C), 40 μl were boiled at 95 °C for 10 min while shaking at 800 r.p.m. on a Thermoshaker (Eppendorf). Protein extracts were then processed using the single-pot solid-phase enhanced sample preparation (SP3). The SP3 protein purification, digest and peptide clean-up were performed manually using carboxylate-modified magnetic particles (GE Life Sciences, GE65152105050250, GE45152105050250)[60]. Beads were conditioned following manufacturer instructions, consisting of 3 washes with water at a concentration of 1 μg μl$^{-1}$. Samples were diluted with 100% ethanol to a final concentration of 50% ethanol. Washed beads (5 μg) were added to each sample, and samples were incubated for 30 min at room temperature and 800 r.p.m. on a Thermoshaker (Eppendorf). Beads were collected on a magnetic rack and washed three times with 80% ethanol, each time incubating the beads and wash solution for 3 min at room temperature and 800 r.p.m. Washed beads were resuspended in 105 μl 50 mM triethylammonium bicarbonate, and 500 ng of sequencing-grade trypsin (Promega) were added for overnight incubation at 37 °C. Supernatants containing peptides were collected the next day and combined with the supernatant of an additional bead wash carried out in 90 μl MilliQ water with sonication for 15 min at room temperature. The samples were dried to completeness and resolubilized in 20 μl of MS sample buffer (3% acetonitrile, 0.1% formic acid) containing iRT peptides (Biognosys). Peptide concentration was determined using the Lunatic UV/Vis polychromatic spectrophotometer (Unchained Labs).

## Liquid chromatography for label-free proteomics quantification

LC–MS/MS analysis was performed on an Orbitrap Fusion Lumos (Thermo Scientific) equipped with a Digital PicoView source (New Objective) and coupled to an M-Class UPLC (Waters). Solvent composition of the two channels was 0.1% formic acid for channel A and 99.9% acetonitrile in 0.1% formic acid for channel B. Column temperature

was 50 °C. Samples were diluted 1:10, and for each sample peptides corresponding to an absorbance of 0.27 were loaded on a commercial ACQUITY UPLC M-Class Symmetry C18 Trap column (100 Å, 5 μm, 180 μm × 20 mm, Waters) connected to an ACQUITY UPLC M-Class HSS T3 column (100 Å, 1.8 μm, 75 μm × 250 mm, Waters). The peptides were eluted at a flow rate of 300 nl min⁻¹. After a 3 min initial hold at 5% B, a gradient from 5 to 22% B in 80 min and 22 to 32% B in additional 10 min was applied. The column was cleaned after the run by increasing to 95% B and holding 95% B for 10 min before re-establishing loading condition.

Samples were measured in randomized order. For the analysis of the individual samples, the mass spectrometer was operated in data-independent mode (DIA). DIA scans covered a range of 396–956 $m/z$ in windows of 8 $m/z$. The resolution of the DIA windows was set to 15,000, with an AGC target value of 500,000, a maximum injection time set to 22 ms and a fixed normalized collision energy of 33%. Each instrument cycle was completed by a full MS scan, monitoring 396–1,000 $m/z$ at a resolution of 60,000. The mass spectrometry proteomics data were handled using the local laboratory information management system[61].

### Proteome quantification

The acquired MS raw data were processed for identification and quantification using FragPipe (v.18.0), MSFragger (v.3.5) and Philosopher (v.4.4.0)[62]. Spectra were searched against the uniprot reference proteome of *Y. entomophaga* downloaded from https://www.uniprot.org/proteomes/UP000266744 on 24 May 2023, concatenated to its reversed decoy database and common protein contaminants. MSFragger-DIA mode for direct identification of peptides from DIA data was used. Strict trypsin digestion was set to a maximum of two missed cleavages. Carbamidomethylation of cysteine was selected as a fixed modification, while methionine oxidation and N-terminal protein acetylation were set as variable modifications. EasyPQP was used to generate a DIA-NN-compatible spectral library. Subsequent quantification was performed with DIA-NN v.1.8.2.

### DEA with PROLFQUA

The R package prolfqua[63] was used to analyse differential expression and to determine group differences, confidence intervals and false discovery rates for all quantifiable proteins. Starting with the precursor abundances reported by DIA-NN, we determined protein abundances using the Tukeys-median polish. Furthermore, we transformed the protein abundances using the variance stabilizing normalization[64].

Since the experiment had two factors, cell type and temperature, we fitted a linear model with two explanatory variables and an interaction term. We examined the following contrasts: LC_37C_vs_LC_25C and WT_37C_vs_WT_25C.

### Intoxication bioassay

*Y. entomophaga* WT, ΔYenTc (*Δchi1/ΔyenA1/ΔyenA2/Δchi2/ΔyenB/ΔyenC1/ΔyenC2*) and *chi2-sfGFP* 3 ml overnight cultures were grown in LB at 30 °C at 250 r.p.m. in a Raytek orbital incubator. Of the overnight culture, 500 μl was used to inoculate 50 ml LB broth, then grown for 18 h at 25 °C. Bacterial debris was removed by centrifugation (10 min; 10,000g, 4 °C), followed by filter sterilization of the supernatant through a 0.2 μm Sartorius Minisart filter into a sterile tube. Of the filtrate, 5 μl was inoculated onto a diced 3 mm³ carrot cube placed on a tray, where healthy third-instar larvae of the New Zealand grass grub (*C. giveni*) were placed. Bioassays were randomized, with a total of 12 larvae per treatment. The negative control comprised a 5 μl aliquot of LB broth and the positive control comprised 5 μl of YenTc control supernatant. The assay was maintained at 15 °C and monitored at day 6 post challenge. Three independent bioassays were undertaken.

Analysis of the data was carried out with statistical software Minitab v.16.2. Data on diseased (%) larvae were compared between treatments using a generalized linear model (GLM) with binomial distributions through a logit link function. The GLM consisted of treatment factor only.

### Mechanical cryo-milling of ΔLC cell pellets

*Y. entomophaga* ΔLC cultures were grown for 24 h at 25 °C with 200 r.p.m. shaking. Of the bacterial culture, 500 ml was then concentrated by centrifugation (3,949g, 4 °C, 10 min). The resulting cell pellet was resuspended in 25 ml of buffer containing 25 mM Tris, 150 mM NaCl pH 7.5 and ¾ tablet of protease inhibitor cocktail (Roche), and directly processed for liquid nitrogen freezing. Here, the bacterial suspension was stepwise pipetted into a falcon tube placed in liquid N₂, resulting in frozen cell pellet drops. The cell pellet drops were stored in liquid N₂ until milling.

Cryo-milling was performed using the SPEX Sample Prep 6870 large freezer/mill (Thermo Fisher, SPEX SamplePrep). Milling of the cells was performed by grinding the cells for 6 cycles at a rate of 15 cycles per second (30 impacts per second) and a run time of 3 min. The sample was cooled between cycles and the coil was inactive for 2 min to cool the sample down. All steps were performed at cryogenic temperatures. The resulting cryo-milled cell powder was directly stored in a falcon tube in liquid N₂.

Freezing of the cell powder was performed by collecting a spatula of cell powder into a reaction tube in liquid N₂. Seconds before application on a glow-discharged cryoEM grid, 100 μl to 1 ml of sample buffer was added.

### Light microscopy

For sample preparation, 1.5% agar pads were freshly poured into a 35-mm-high glass-bottom μ-dish (ibidi). After 30 min, the agar pad was inverted and five 1.5 μl drops of respective bacterial culture were equally spotted onto the pad. The bacterial cultures were adjusted to an OD₆₀₀ of 0.05 for fLM experiments and to an OD₆₀₀ of 0.01 for timelapse imaging. After drying, the agar pad was mounted on a 35-mm-high glass-bottom μ-dish (ibidi) supplemented with a wet Kim wipe and closed with parafilm and vacuum grease.

Images were recorded using a ×100 phase-contrast objective on a Leica Thunder Imager 3D Cell Culture equipped with a Leica DFC9000 GTC CMOS camera (2,048 × 2,048 pixels, pixel size 6.5 mm) at a stage temperature of 25 °C. The Leica Application Suite X (LAS X) software platform was used for acquisition and the resulting images were analysed using Fiji[65], GraphPad Prism and Microsoft Excel. Different cell types were quantified using the cell counter function in Fiji. For timelapse imaging of *Y. entomophaga chi2-sfGFP*, single images were recorded every 10, 15 or 20 min over several hours using a high-speed software autofocus with a local range of 25 μm using the bright-field channel.

### Plunge freezing for cryoET imaging

*Y. entomophaga* WT and mutant cultures were concentrated for bacterial lawn preparations by centrifugation to reach a final OD₆₀₀ of 30. A volume of 3.5 μl of cell suspension or a spatula of cryo-milled ΔLC cell powder diluted in 1 ml of buffer were applied to glow-discharged copper EM grids (R2/1 or R2/2, Quantifoil) and subsequently blotted two times and plunged into liquid ethane/propane[66] with a blot force of 0 using a Vitrobot Mark IV (Thermo Fisher)[67]. Using a Teflon sheet on one side, all samples were blotted exclusively from the back[68] for 4–6 s after a waiting time of 1 min. Purified YenTc particles were plunge frozen as described before[13]. Frozen grids were stored in liquid nitrogen until loaded into the microscope.

### CryoFIB milling

Automated sequential cryoFIB milling was performed as previously described[26,69,70]. Plunge-frozen grids were clipped into FIB-autoloader grids (Thermo Fisher) and loaded into a 40° pre-tilted scanning electron

microscopy (SEM) holder (Leica Microsystems)[71]. For grid transfer, a VCT500 cryo-transfer system (Leica Microsystems) was used and grids were sputter-coated with a 4 nm tungsten layer using an ACE600 cryo-sputter coater (Leica Microsystems). After grid transfer to a Crossbeam 550 FIB-SEM dual-beam instrument (Carl Zeiss) equipped with a copper-band cooled mechanical cryo-stage (Leica Microsystems), the gas injection system was used to deposit an organometallic platinum precursor layer onto each grid. Identification of suitable targets was done by SEM imaging (3 kV, 58 pA), and milling patterns were placed onto the ROI's FIB image (20 pA, 30 kV) using the SmartFIB software. Lamella width was set between 8 and 10 µm with a target thickness of 200 nm. In total, four FIB currents were used, gradually reducing with lamella thickness from 700 pA, 300 pA and 100 pA, to 50 pA for final polishing. The grids with the prepared lamella were unloaded and stored in liquid nitrogen.

### Correlative light and electron microscopy

CryoFIB-milled lamellae on EM grids were imaged using a Zeiss LSM900 equipped with Airyscan2 detector and a Linkam CMS196V3 cryo-stage in a de-humidified room (humidity <15%). To localize lamellae, EM grid overview images were acquired with a ×5/0.2C Epiplan-Apochromat objective. Z-stacks of each lamella were collected with a ×100/0.75 DIC LD EC Epiplan-Neofluar objective. Z-stacks were recorded using a confocal track detecting Chi2-sfGFP, transmitted light and reflected light, and a separate Airyscan track detecting Chi2-sfGFP. Confocal imaging stacks were deconvolved using the Zeiss LSM Plus processing function, and Airyscan data were processed with Zeiss joint deconvolution (jDCV, 20 iterations). Maximum intensity projections were created using the extended depth of focus function in Zen Blue (Carl Zeiss, v.3.5) software.

CryoLM data were then further used to guide cryoET data collection on lamellae in X/Y dimensions[72]. CryoLM images were aligned to corresponding EM overview images of lamellae using prominent landmarks and cell shapes. After cryoET data collection, low-magnification EM lamella overview images were converted to tiff files using 'mrc2tif' in IMOD[73,74] before alignment. For precise correlation of cryoLM and cryoEM lamella overview images, cryoEM tiff files were then imported into ZEN Connect (within Zen Blue, Carl Zeiss, v.3.5) and precisely correlated with the cryoLM maximum intensity projections using the Point Alignment Wizard (allowing only for translation and rotation). The correlated images were exported from Zen Blue and visualized in Fiji[65].

### CryoET

CryoET data were recorded on Titan Krios G3 and G4 (Thermo Fisher) microscopes operating at 300 kV and equipped either with a Quantum LS imaging filter or a BioContinuum imaging filter (slit width 20 eV), combined with a K3 direct electron detector (Gatan). All data were acquired using SerialEM[75,76]. Tilt series on lamellae were either recorded in a bidirectional tilt scheme using a custom-made SerialEM script or in a dose-symmetric tilt scheme using PACE tomo[77] with 3° angular increments. Besides tilt series used for filament averaging, which have been acquired with a pixel size of 2.68 Å at specimen level and a defocus range of −3 to −5 µm, all tilt series on lamellae were recorded with a pixel size of 4.51 Å and a defocus of −8 µm. Tilt series on cell lysate or purified YenTc-sfGFP particles were acquired with a bidirectional tilt scheme using 2° angular increments and with a pixel size of 4.51 Å and a defocus of −8 µm. Data on YenTc (WT) particles were collected similarly but with a pixel size of 2.68 Å. All tilt series covered 120° of angular range and had a cumulative electron dose of 140–160 e⁻ Å⁻².

### Cryo-tomogram reconstruction, data processing and segmentation

Tilt series were motion-corrected using 'alignframes' in the IMOD package[73,74] and 2x- or 4x-binned cryo-tomograms were reconstructed by weighted back projection in IMOD. Automated cryoET alignment and reconstruction were done using AreTomo[78] and denoising using cryoCARE for initial data screening. A set of in-house scripts[79] were used to facilitate batch processing. Contrast transfer function (CTF) was determined using the GCTF software[80]. More precise tilt-series alignments and reconstructions were obtained using the IMOD package. Cryo-tomograms were filtered using IsoNet to improve contrast for visualization, particle picking and template matching[81]. Segmentations shown were generated using Dragonfly (Object Research Systems; www.theobjects.com/dragonfly) as previously described[82]. A 5-class U-Net with 2D input was trained on 5–15 IsoNet[81] tomogram slices to detect background voxels, cytoplasmic and outer membranes of lysing and intact cells. Cells and cell types were quantified by examining high-quality tomograms, and graphs were prepared using GraphPad Prism and Microsoft Excel. All segmentations produced by the neural network were cleaned up in Dragonfly, subsequently exported as binary tiff and converted to mrc files using 'tif2mrc' in IMOD[73,74]. Segmentations were imported into ChimeraX[83,84], Gaussian filtered and processed with the surface smoothing function. All segmentations were visualized using ChimeraX. For visualization purposes, subtomogram averaged YenTc or YenTc-sfGFP particles were projected into the segmented model of the used tomograms of particle picking using ArtiaX[85] in ChimeraX.

### Subtomogram averaging of purified YenTc

Purified YenTc and YenTc-sfGFP particles were manually picked in individual cryo-tomograms (n = 681 particles from 11 tomograms and n = 739 particles from 3 tomograms, respectively) using a dipole to specify their long axis. Particle extraction, alignment and averaging were done using Dynamo[86] in Matlab. For YenTc (WT), 4x-binned particles were aligned for 3 iterations, with a default spherical mask and a box size of 68 × 68 × 68 pixels. Afterwards, the dataset was split in half and independently aligned for another 9 iterations, applying additional five-fold symmetry. The datasets were cleaned according to cross-correlation (CC) values (CC cut-off: 0.34) and subsequently aligned for 3 iterations with a tight mask. For YenTc-sfGFP, 4x-binned particles were aligned for 3 iterations, with a default spherical mask and a box size of 40 × 40 × 40 pixels. The dataset was split in half and independently aligned for another 8 iterations, applying additional five-fold symmetry using 2x-binned particles with a box size of 80 × 80 × 80 pixels. The particles were cleaned according to their CC value (CC cut-off: 0.27) and subsequently aligned for 4 iterations with a tight mask. The final, five-fold symmetrized 3D reconstructions were calculated from 432 YenTc (WT) particles and 412 YenTc-sfGFP particles. Gold-standard Fourier shell correlation curves were determined from the half-maps using Dynamo. 3D rendering was done with UCSF ChimeraX[83,84]. For the calculation of a difference map, the 3D reconstructions were low-pass filtered to 50 Å resolution, and box and pixel sizes were unified using 'relion_image_handler' tool to a pixel size of 9.02 Å with a box size of 80 × 80 × 80 pixels. The difference map was created using the DIFF-MAP software package from the Grigorieff lab (https://grigorieflab.umassmed.edu/diffmap).

### Subtomogram averaging of filament particles

All preprocessing was done as described above (Extended Data Fig. 10). Filament particle picking and initial subtomogram averaging attempts were done using Dynamo software[86]. Thus, 13,104 particles were manually picked from the 4x-binned tomograms collected at −8 µm defocus and a pixel size of 4.51 Å per pixel, followed by extraction with a box size of 48 × 48 × 48 pixels with 8x binning. A cylinder of the filament diameter was used as an initial 3D reference and 3 rounds of refinements were performed to obtain a reliable initial model. To obtain a higher-resolution 3D reconstruction, another dataset was used with lower target defocus range (−3 to −5 µm) and higher magnification with a pixel size of 2.68 Å per pixel (see Extended Data Fig. 10). Particles (35,332 4x-binned) were manually picked from 42 denoised

tomograms using Dynamo. Particles were then extracted from the original tomograms and further processed in Relion4 (ref. 87). Initial 3D refinements, followed by 3D classification, were performed at 4x and 2x binning. Final rounds of 3D refinements, CTF refinements and frame alignments were performed at the original sampling. A total of 23,158 particles contributed to the final 3D reconstruction after focusing on a single 3D reconstruction, revealed by 3D classification. No helical symmetry (except for the illustration purpose in Fig. 5b) was applied during the data processing. Helical parameters for illustration purposes were determined by translation and rotation of the final map in Chimera, followed by refinement and validation of the results using 'relion_helix_toolbox'.

### Docking the atomic models of the protein candidates into the cryoEM map

We used the Chimera Segmentation tool to select a single asymmetric unit. Alphafold2-predicted atomic models of 20 candidates were manually analysed in Chimera for a possibility of potential fit into the asymmetric unit of the resulting cryoEM map at 10 Å resolution. All candidates appeared too small to occupy a single asymmetric unit, therefore multiple copies of the same candidates were checked. One homotetramer (M66/StcE) appeared to fit properly. To extend the search for other potential fitting options and identify which protein composes the filament, we used Colores tool[40,41] in Situs 3.1, applying exhaustive rigid-body search on a discrete 6D lattice of the Alphafold2-predicted structures of the protein candidates.

### Template matching

Template matching and subsequent subtomogram averaging were performed using Dynamo[86]. The previously calculated YenTc-sfGFP 3D reconstruction was used as initial template for template matching in 4x-binned (pixel size: 18.04 Å) and IsoNet-corrected[81] tomograms. Following template matching, particles were extracted according to their CC values (CC cut-off for YenTc (WT): 0.2; CC cut-off for YenTc-sfGFP: 0.17), resulting in 1,555 particles for YenTc (WT) and 1,133 particles for YenTc-sfGFP. Particles were visually inspected to sort out false picks, which resulted in 304 and 567 final particles for YenTc (WT) and YenTc-sfGFP, respectively (see Extended Data Fig. 5). The resulting particle coordinates from the cleaned template matching dataset were used to extract particles from non-filtered 4x-binned tomograms and subjected to subtomogram averaging (box size 40 × 40 × 40 pixels). As initial reference for all subsequent subtomogram averaging, the previously calculated YenTc-sfGFP 3D reconstruction was low-pass filtered to 200 Å resolution. After 6 global alignment iterations, datasets were split in half and independently aligned for another 6 alignment iterations. The final 3D reconstructions were not CC cleaned and represent all hits from template matching. Fourier shell correlation calculations were performed in Dynamo using half-maps.

For visualization purposes, subtomogram averaged YenTc or YenTc-sfGFP particles were projected into the segmented model of the used tomograms of particle picking using ArtiaX[85] in ChimeraX.

### Reporting summary

Further information on research design is available in the Nature Portfolio Reporting Summary linked to this article.

### Data availability

Example cryo-tomograms (EMD-18953, EMD-18954, EMD-18955, EMD-18957, EMD-18958, EMD-18960, EMD-18961, EMD-18962, EMD-19370–EMD-19381) and subtomogram averages (EMD-18970–EMD-18972) have been deposited in the Electron Microscopy Data Bank (EMDB). All relevant proteomic data have been deposited to the ProteomeXchange Consortium via the PRIDE partner repository (http://www.ebi.ac.uk/pride) with the dataset identifier PXD048008. Source data are provided with this paper.

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

## Acknowledgements

We thank ScopeM for instrument access at ETH Zürich; J. Xu for feedback on generating the M66/StcE filament structure; D. Böhringer for discussions and comments on the manuscript; the Functional Genomics Center Zürich, P. Nanni, W. Wolski and L. Kunz for mass spectrometry services and support. The Pilhofer lab was supported by the European Research Council (679209), the Swiss National Science Foundation (310030_212592) and the NOMIS foundation. M.H. was supported by Ministry of Business, Innovation and Employment C10X1805.

## Author contributions

M.F. performed sample preparations, western blotting, fLM data collection and processing, cryoFIB milling, cryoET data collection and processing, segmentation, subtomogram averaging, template matching, CLEM, filament particle picking and mass spectrum analyses for potential candidates. P.A. solved and analysed the filament structure, and performed docking of the candidate proteins. C.F.E. engineered the *chi2-sfGFP* mutant. Y.-W.L. engineered the ΔLCΔYenTc and ΔLCΔM66 mutant. G.L.W. supported subtomogram averaging, template matching and writing. F.W. supported the CLEM approach and Dragonfly segmentation. M.H. performed the toxicity assay and provided feedback on the project. M.S. provided the ΔLC mutant. M.F. and M.P. wrote the manuscript with input from all authors.

## FundingInformation

## Competing interests

The authors declare no competing interests.

## Additional information

**Extended data** is available for this paper at https://doi.org/10.1038/s41564-024-01611-2.

**Correspondence and requests for materials** should be addressed to Martin Pilhofer.

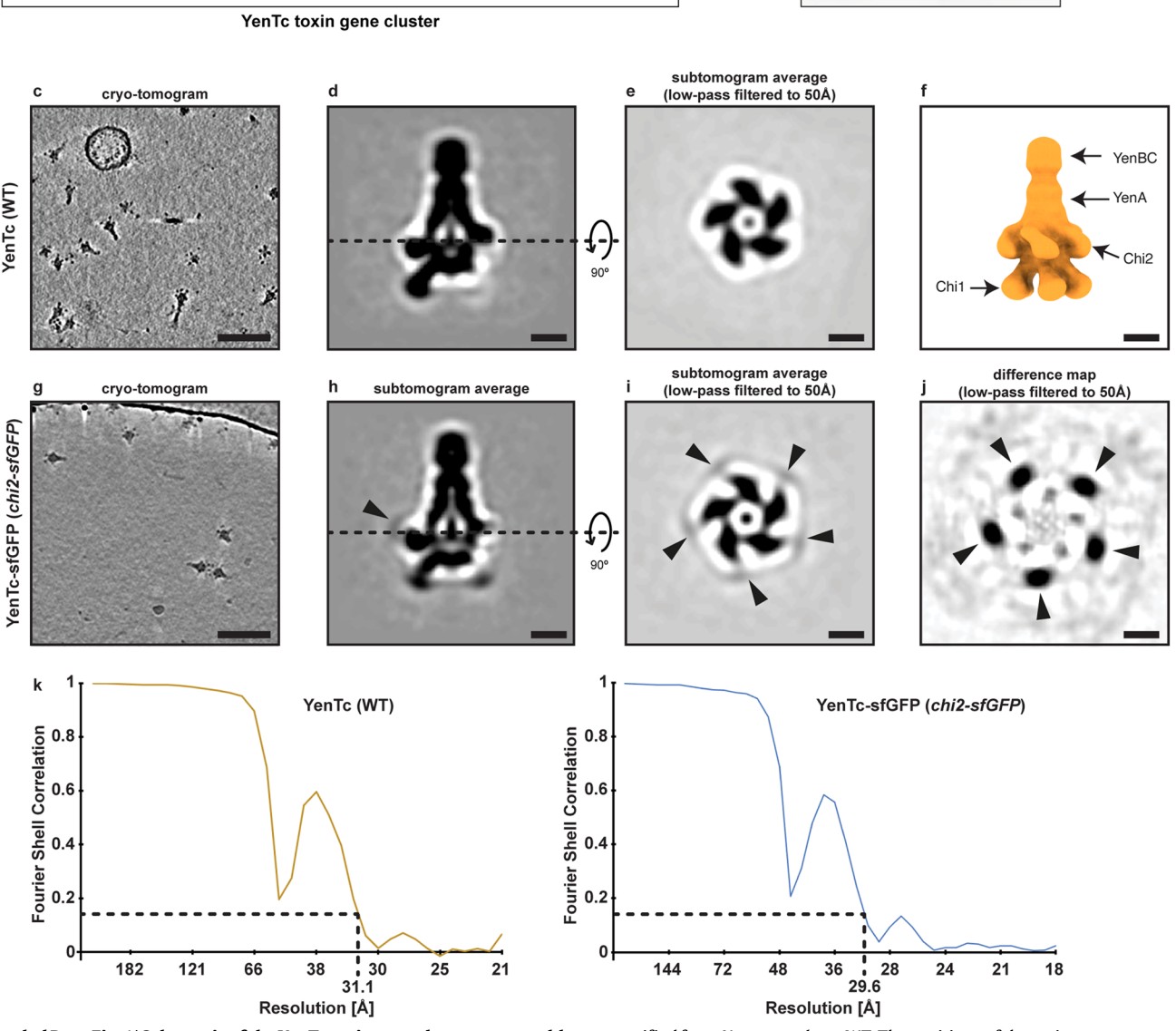

**Extended Data Fig. 1 | Schematic of the YenTc toxin gene cluster, western blot and subtomogram averages of YenTc particles purified from _Y. entomophaga_ WT and _chi2-sfGFP_. a**: Shown is a schematic of the YenTc toxin gene cluster in _Y. entomophaga_ (gene names are indicated). **b**: Shown is a western blot against GFP and RecA (loading control) of _Y. entomophaga_ WT and a _chi2-sfGFP_ mutant. GFP could only be detected in the cell pellet and supernatant (sup.) of the mutant and the observed bands correspond to the size of Chi2-sfGFP (~100 kDa). The experiment was repeated three times with similar results. **c**: Slices through cryo-tomograms of a YenTc preparation from _Y. entomophaga_ WT (top, 1.1 nm thick slice) and _chi2-sfGFP_ (bottom, 0.9 nm thick slice). Bar: 100 nm. **d/e**: Shown are longitudinal (d) and perpendicular (e) sections through five-fold rotationally symmetrized subtomogram average of YenTc from _Y. entomophaga_ WT. Slice thickness: 1.1 nm. Bars: 10 nm. **f**: Subtomogram average of YenTc holotoxin

purified from _Y. entomophaga_ WT. The positions of the main components of YenTc are indicated. **g**: Slice through cryo-tomogram of a YenTc preparation from _Y. entomophaga chi2-sfGFP_. Slice thickness: 0.9 nm. Bar: 100 nm. **h/i**: Shown are longitudinal (h) and perpendicular (i) sections through five-fold rotationally symmetrized subtomogram average of YenTc from _Y. entomophaga chi2-sfGFP_. Slice thickness: 0.9 nm. Note the additional density, potentially accounting for sfGFP (black arrowhead) in the average. Bars: 10 nm. **j**: Difference map between subtomogram averages of YenTc (_Y. entomophaga_ WT) and YenTc-sfGFP (_Y. entomophaga chi2-sfGFP_) revealed that the only major differences are additional densities in the periphery of the Chi2 domain, potentially accounting for sfGFP. Slice thickness: 0.9 nm. Bar: 10 nm. **k**: Fourier Shell Correlation (FSC) analyses of half-datasets of both subtomogram averages indicated resolutions of ~31.1 Å and ~29.6 Å, respectively.

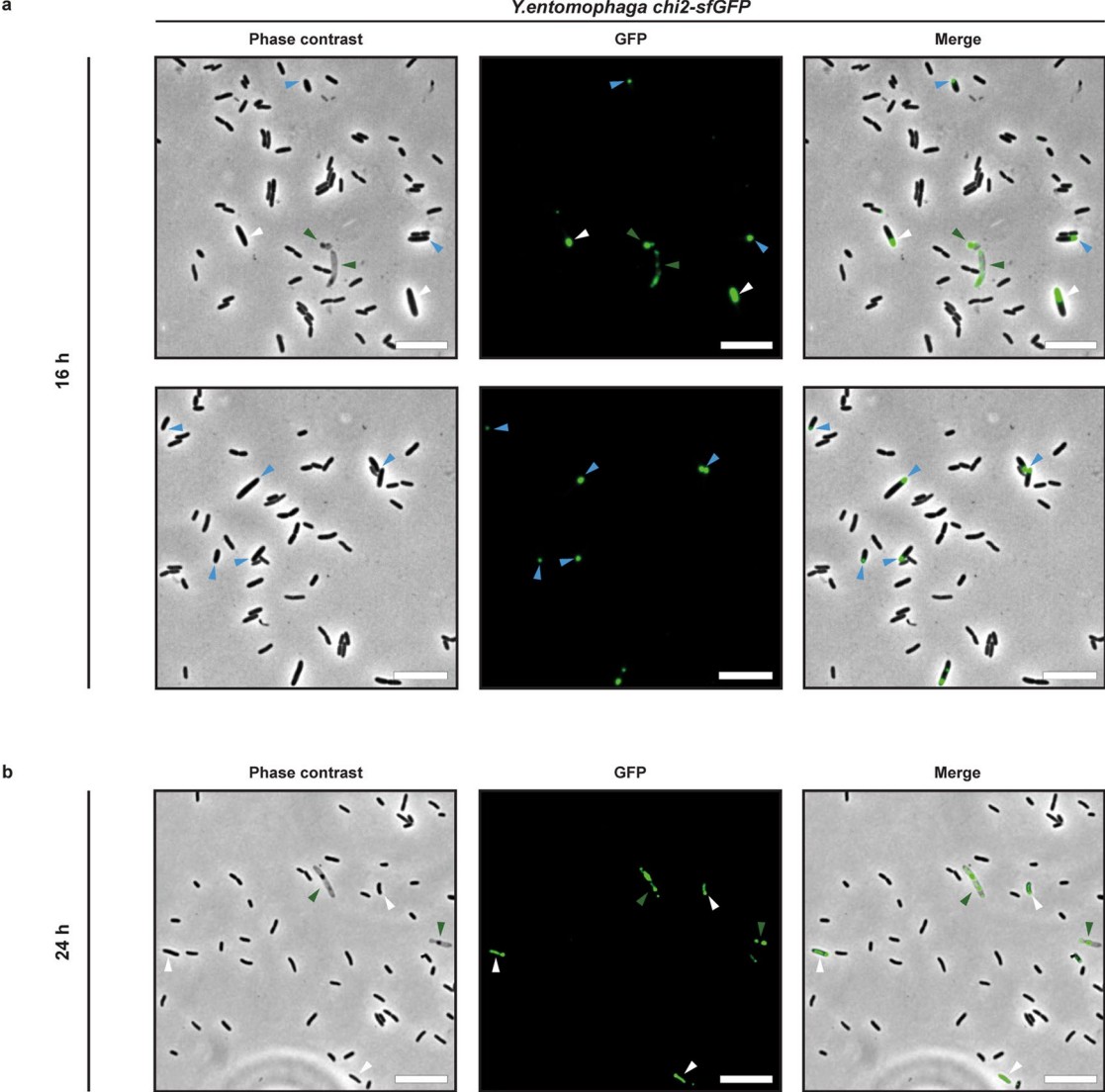

**Extended Data Fig. 2 | fLM images of *Y. entomophaga chi2-sfGFP* at different time points shows that only a subpopulation of cells expresses YenTc-sfGFP.** fLM of *Y. entomophaga chi2-sfGFP* grown for 16 h **(a)** or 24 h **(b)** revealed the expression of Chi2-sfGFP only in a subpopulation of cells. Foci at the cell poles (blue arrowheads) as well as distributed signal (white arrowheads) were found in somewhat elongated and thicker cells. Note that lysing cells (loss of phase contrast, green arrowheads) show sfGFP signal. The experiment was repeated three times for the time point 16 h, and four times for the time point 24 h with $n_{total, 16h}$ = 4810 and $n_{total, 24h}$ = 7317 ($n$ > 1000 cells for each experiment). Bar: 10 μm.

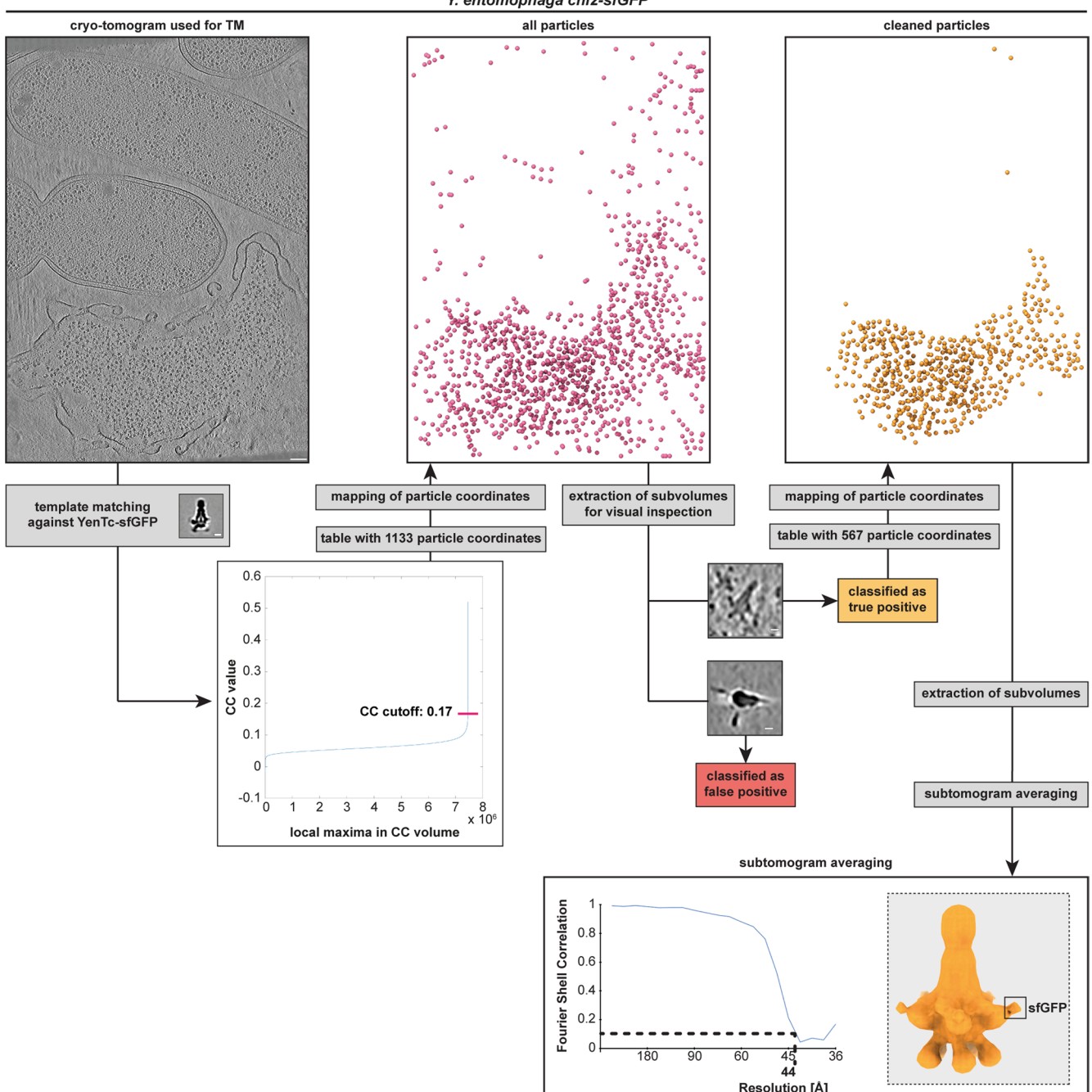

**Extended Data Fig. 3 | Workflow of YenTc template matching of**
***Y. entomophaga chi2-sfGFP.*** Workflow of template matching shown in Fig. 2g/h.
The shown tomogram was used for template matching against the previously
calculated YenTc-sfGFP subtomogram average (Extended Data Fig. 1). After
initial template matching, particles were selected according to their CC-values
(CC-cutoff: 0.17), which resulted in 1,133 particles (projected onto tomogram of
ROI, middle, magenta). After visual inspection of the individual particles, 567
particles were classified as true positives (projected onto ROI, right, orange).
Particle coordinates were then used for subtomogram averaging, which resulted
in a structure resembling YenTc-sfGFP (orange 3D volume in grey box). Note the
additional density on the Chi2 domain, likely accounting for sfGFP. Fourier Shell
Correlation (FSC) analysis of two half-datasets of this subtomogram average
resulted in a resolution of ~44 Å.

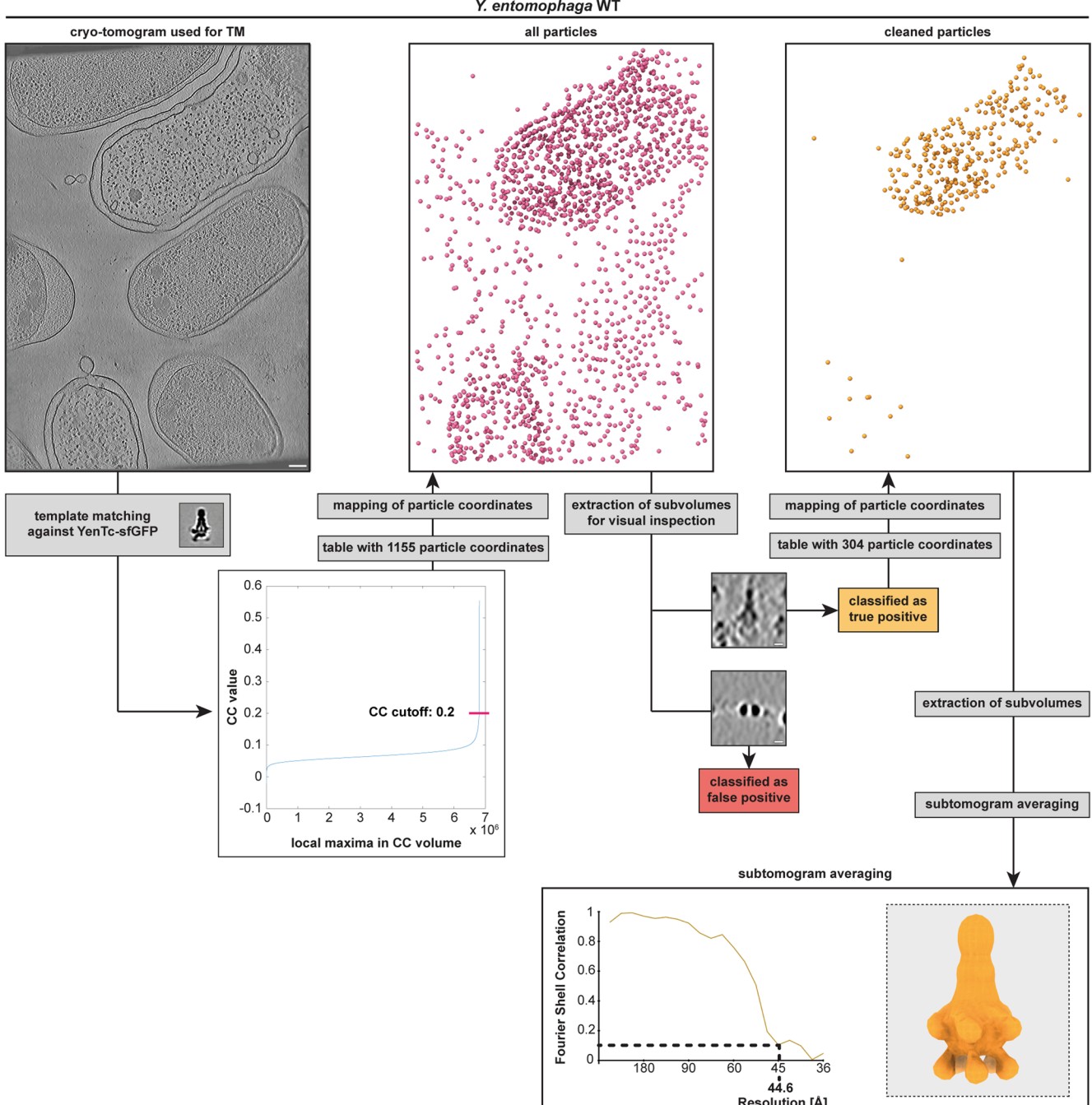

**Extended Data Fig. 4 | Workflow of YenTc template matching of**
***Y. entomophaga* WT.** Workflow of template matching shown in Fig. 3f/g.
The shown tomogram was used for template matching against the previously
calculated YenTc-sfGFP subtomogram average (Extended Data Fig. 1). After
initial template matching, particles were selected according to their CC-values
(CC-cutoff: 0.2), which resulted in 1,555 particles (projected onto tomogram of

ROI, middle, magenta). After visual inspection of the individual particles, 304
particles were classified as true positives (projected onto ROI, right, orange).
Particle coordinates were then used for subtomogram averaging, which resulted
in a structure resembling YenTc (orange 3D volume in grey box). Fourier Shell
Correlation (FSC) analysis of two half-datasets of this subtomogram average
resulted in a resolution of ~44.6.

**Extended Data Fig. 5 | Lysing cells contain YenTc particles that do not directly interact with the cytoplasmic membrane.** Shown are more examples of cryo-tomograms of lysing *Y. entomophaga chi2-sfGFP* (**a**) and WT (**b**) cells. Cells typically contained large numbers of YenTc particles that do not directly interact with the cytoplasmic membrane (CM) but are instead separated from the CM by a low density region. Importantly, these cells frequently showed transenvelope-spanning structures (shown in close-up views in insets) that spanned between CM and outer membrane (OM) in particular at construction sites (arrowheads). Slice thickness: 18.04 nm. Bar: 100 nm.

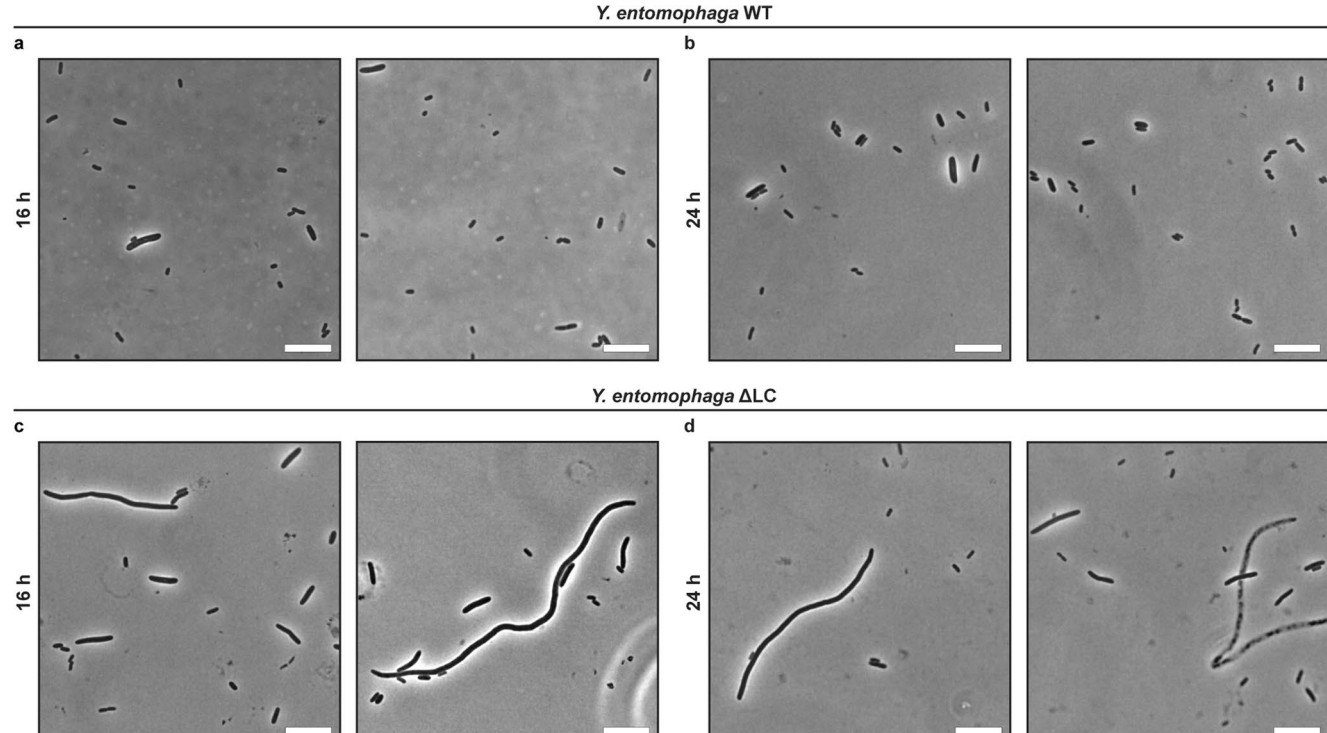

**Extended Data Fig. 6 | Phase contrast LM images of *Y. entomophaga* WT and ΔLC show distinct cell shapes. a/b**: fLM of *Y. entomophaga* WT grown for 16 h (a) or 24 h (b) showed rod-shaped cells up to 10 µm in length. Note that some cells are thicker and have more contrast in brightfield, similar to observations made with GFP-positive cells in *Y. entomophaga chi2-sfGFP*. The experiment was repeated six times with similar results. Bar: 10 µm. **c/d**: fLM of *Y. entomophaga* ΔLC cells grown for 16 h (c) or 24 h (d). Besides the observation of WT-like cell shapes, some cells had a larger diameter and were substantially elongated, reaching up to 80 µm in length. The experiment was repeated five times with similar results. Bar: 10 µm.

a

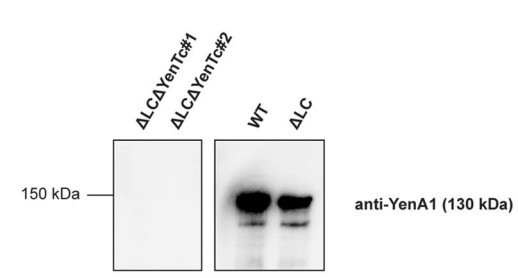

b

*Y. entomophaga* ΔLCΔYenTc

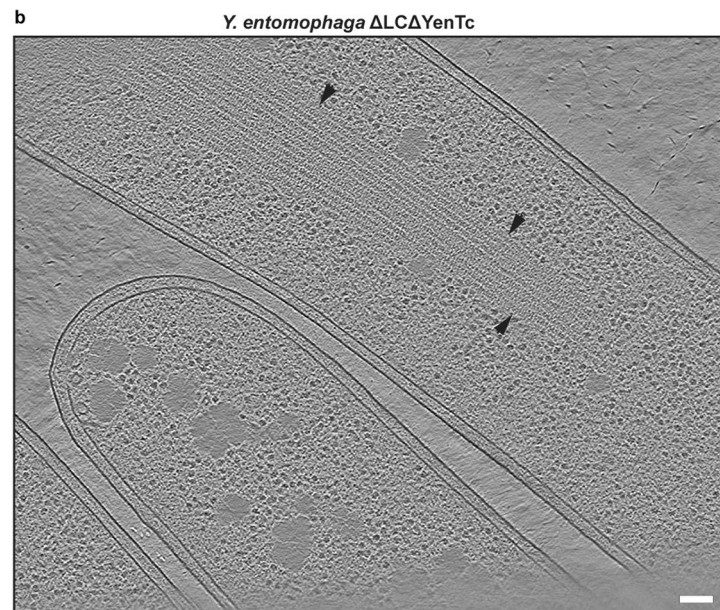

**Extended Data Fig. 7 | Cytoplasmic filaments in *Y. entomophaga* are not composed of YenTc components. a**: Western blot analysis of cell pellets of *Y. entomophaga* ΔLCΔYenTc, WT and ΔLC strains confirmed the absence of YenA1 in the ΔLCΔYenTc mutant. The experiment was repeated two times with similar results. **b**: Cryo-tomogram of *Y. entomophaga* ΔLCΔYenTc showed distinct filament bundles (arrowheads), excluding the possibility that the filaments are composed of YenTc components. Slice thickness: 18.04 nm. Bar: 100 nm.

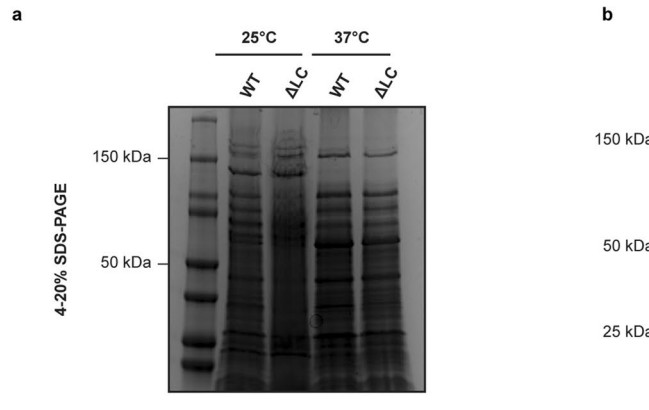

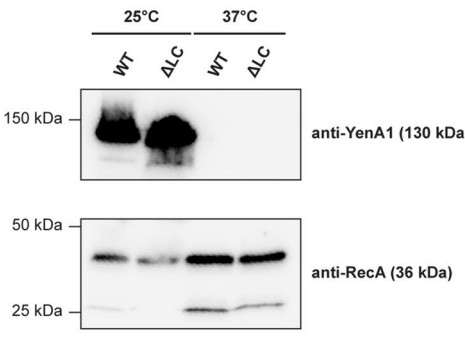

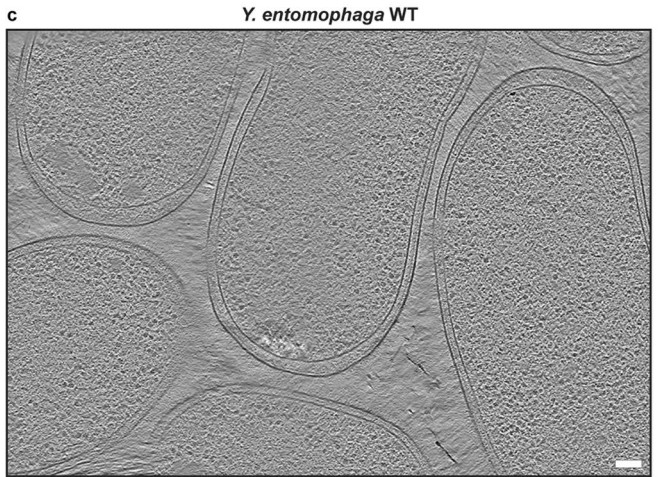

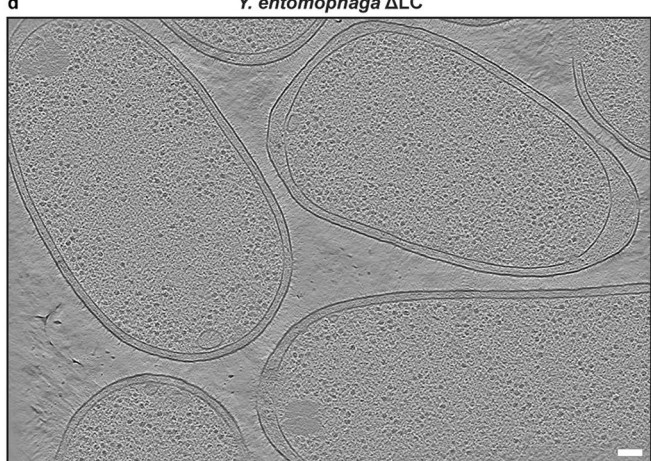

**Extended Data Fig. 8 | Increase of temperature leads to absence of YenTc and filaments. a**: SDS-PAGE analysis of cell lysates of *Y. entomophaga* WT and ΔLC mutant grown at 25 °C and 37 °C. Characteristic protein bands for YenTc were absent at 37 °C. The experiment was repeated three times with similar results. **b**: Western blot analysis of cell lysates of *Y. entomophaga* WT and ΔLC mutant grown at 25 °C and 37 °C. Cytoplasmic expression of YenA1 was absent at 37 °C. The experiment was repeated three times with similar results. **c/d**: Slices through cryo-tomograms of *Y. entomophaga* WT (c) and ΔLC (d) cells showed no apparent YenTc densities and the absence of elaborate filament bundles. Slice thickness: 18.04 nm. Bar: 100 nm.

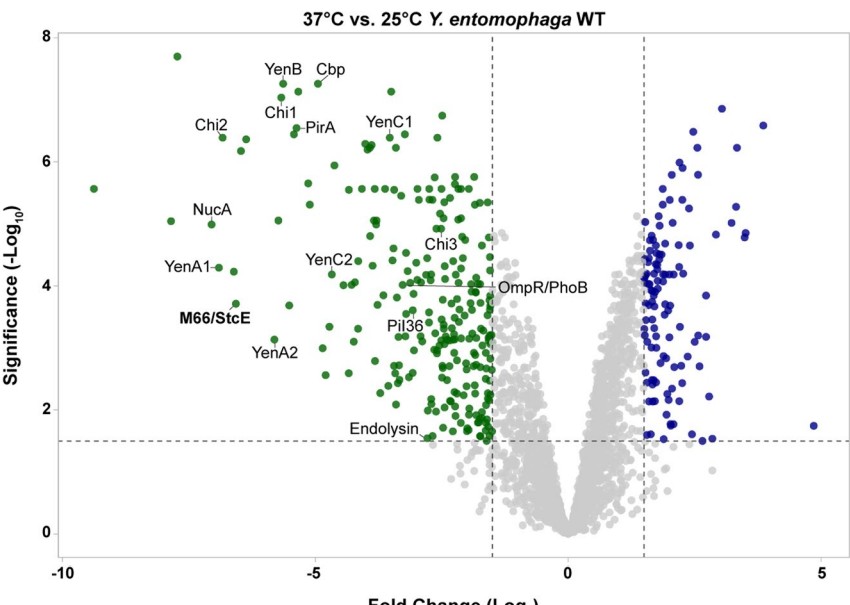

**Extended Data Fig. 9 | Virulence factors and endolysin are highly upregulated at 25 °C in WT cells.** *Y. entomophaga* WT cells were grown at 37 °C or 25 °C (control standard) and analyzed by mass spectrometry and differential expression analysis (DEA). The volcano plot shows the -$\log_{10}$ false discovery rate (FDR) as a function of $\log_2$ fold change (FC) when comparing the WT samples grown at 37 °C with those grown at 25 °C. These results match the findings from ΔLC cells (see Fig. 5a), except that here also the endolysin was found to be upregulated at 25 °C. All negative FC values ($\log_2$ base), their false discovery rate values as well as their confidence intervals can be found in the Supplementary Table 2.

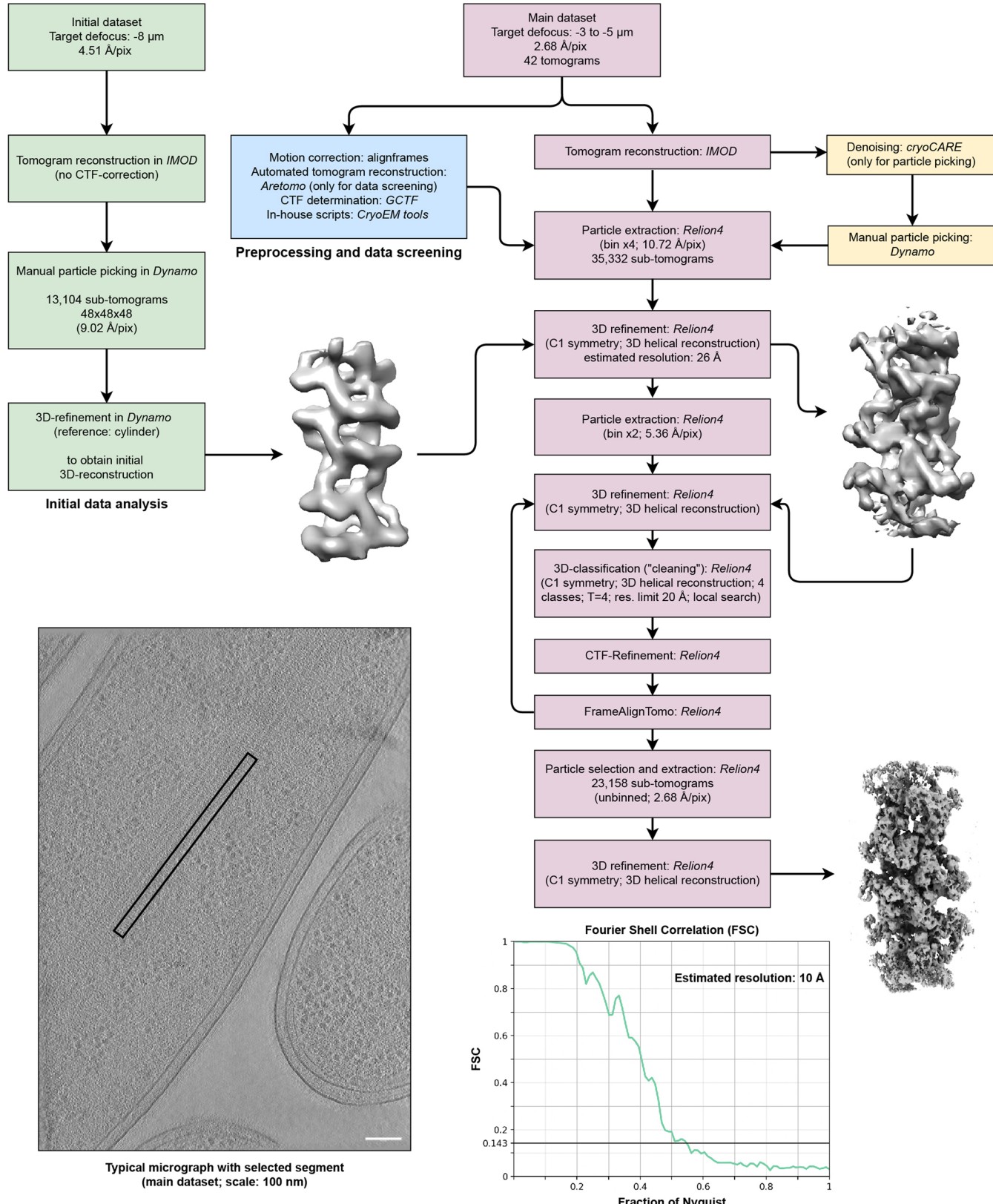

**Extended Data Fig. 10 | Processing pipeline for subtomogram averaging of the M66/StcE filaments.** For details on structural determination see also the Materials and Methods section.

# Reporting Summary

## Statistics

For all statistical analyses, confirm that the following items are present in the figure legend, table legend, main text, or Methods section.

| n/a | Confirmed | |
|---|---|---|
| ☐ | ☒ | The exact sample size (*n*) for each experimental group/condition, given as a discrete number and unit of measurement |
| ☐ | ☒ | A statement on whether measurements were taken from distinct samples or whether the same sample was measured repeatedly |
| ☐ | ☒ | The statistical test(s) used AND whether they are one- or two-sided <br> *Only common tests should be described solely by name; describe more complex techniques in the Methods section.* |
| ☒ | ☐ | A description of all covariates tested |
| ☒ | ☐ | A description of any assumptions or corrections, such as tests of normality and adjustment for multiple comparisons |
| ☐ | ☒ | A full description of the statistical parameters including central tendency (e.g. means) or other basic estimates (e.g. regression coefficient) AND variation (e.g. standard deviation) or associated estimates of uncertainty (e.g. confidence intervals) |
| ☐ | ☒ | For null hypothesis testing, the test statistic (e.g. *F*, *t*, *r*) with confidence intervals, effect sizes, degrees of freedom and *P* value noted <br> *Give P values as exact values whenever suitable.* |
| ☒ | ☐ | For Bayesian analysis, information on the choice of priors and Markov chain Monte Carlo settings |
| ☒ | ☐ | For hierarchical and complex designs, identification of the appropriate level for tests and full reporting of outcomes |
| ☒ | ☐ | Estimates of effect sizes (e.g. Cohen's *d*, Pearson's *r*), indicating how they were calculated |

*Our web collection on statistics for biologists contains articles on many of the points above.*

## Software and code

Policy information about availability of computer code

| | |
|---|---|
| Data collection | cryoET data collection: Serial EM 4.1.0 beta <br> cryoFIB milling: Zeiss SmartFIB 1.14, Zeiss SmartSEM 5.0 <br> cryo light microscopy: ZEN Blue (Carl Zeiss Microscopy) version 3.5 <br> light microscopy: Leica Application Suite X (Las X) version 3.7.6.25997 <br> mass spectrometry: local laboratory information management system (LIMS) |
| Data analysis | cryo light microscopy: ZEN Blue (Carl Zeiss Microscopy) version 3.5, ZEN Connect (within ZenBlue, Carl Zeiss Microscopy) version 3.5, Fiji 2.9.0 <br> mass spectrometry: Scaffold version 5.2.2 (proteome software), local laboratory information management system (LIMS), FragPipe (version 18.0), MSFragger (version 3.5), Philosopher (version 4.4.0), DIA-NN version 1.8.2. <br> cryoET, subtomogram averaging and template matching: IMOD 4.11, MatlabR2022A (9.12.0.1884302), Dynamo version 1.1.532, UCSF Chimera version 1.14, UCSF ChimeraX version 1.6., Relion 4.0, GCTF version 1.06, IsoNet version 0.2, Fiji 2.9.0, AreTomo version 1.3.4, cryoCARE version 0.2.1, DIFFMAP software package 120330, Situs 3.1 (Colores) <br> segmentation: Dragonfly version 2022.2.0.1367, UCSF ChimeraX version 1.6, ArtiaX version 0.1 <br> intoxication assay: Minitab version 16.2 <br> quantification: GraphPad Prism version 9.5.1, Microsoft Excel version 16.78 |

For manuscripts utilizing custom algorithms or software that are central to the research but not yet described in published literature, software must be made available to editors and reviewers. We strongly encourage code deposition in a community repository (e.g. GitHub). See the Nature Portfolio guidelines for submitting code & software for further information.

## Data

Policy information about <u>availability of data</u>

All manuscripts must include a <u>data availability statement</u>. This statement should provide the following information, where applicable:
- Accession codes, unique identifiers, or web links for publicly available datasets
- A description of any restrictions on data availability
- For clinical datasets or third party data, please ensure that the statement adheres to our <u>policy</u>

Example cryo-tomograms (EMD-18953, EMD-18954, EMD-18955, EMD-18957, EMD-18958, EMD-18960, EMD-18961, EMD-18962, EMD-19370 – EMD-19381) and subtomogram averages (EMD-18970 - EMD-18972) are uploaded to EMDB. All relevant proteomic data are deposited to the ProteomeXchange Consortium via the PRIDE (http://www.ebi.ac.uk/pride) partner repository with the data set identifier PXD048008.

Other datasets used in this study from the UniProt database and AlphaFold protein structure database: A0A3S6EXX9, A0A3S6EWV3, A0A3S6EX30, A0A3S6EXC9, A0A3S6EWV9, A0A3S6EWX1, A0A3S6EWW7, A0A3S6EXR6, A0A3S6F1Q8, A0A3S6F569, A0A3S6F007, A0A3S6F4M5, A0A3S6F5G2, A0A3S6F4L4, A0A3S6EYX4.

## Research involving human participants, their data, or biological material

Policy information about studies with <u>human participants or human data</u>. See also policy information about <u>sex, gender (identity/presentation), and sexual orientation</u> and <u>race, ethnicity and racism</u>.

| | |
|---|---|
| Reporting on sex and gender | N/A |
| Reporting on race, ethnicity, or other socially relevant groupings | N/A |
| Population characteristics | N/A |
| Recruitment | N/A |
| Ethics oversight | N/A |

Note that full information on the approval of the study protocol must also be provided in the manuscript.

# Field-specific reporting

Please select the one below that is the best fit for your research. If you are not sure, read the appropriate sections before making your selection.

☒ Life sciences          ☐ Behavioural & social sciences          ☐ Ecological, evolutionary & environmental sciences

For a reference copy of the document with all sections, see <u>nature.com/documents/nr-reporting-summary-flat.pdf</u>

# Life sciences study design

All studies must disclose on these points even when the disclosure is negative.

| | |
|---|---|
| Sample size | subtomogram averaging:<br>81 particles found in 11 high-quality tomograms were used for initial in situ subtomogram average of purified YenTc. Final particle number after cross-correlation cleaning: 432.<br>739 particles found in 3 high-quality tomograms were used for initial subtomogram average of YenTc-Chi2-sfGFP. Final particle number after CC cleaning: 412.<br>35,332 particles found in 42 high-quality tomograms were used for subtomogram average of the filament particles.<br><br>template matching:<br>Particles (n = 1555 for WT, n = 1133 for chi2-sfGFP) were identified via template matching with a CC cutoff value of 0.2 and 0.17. After visual inspection, n = 304 for WT, n = 567 for chi2-sfGFP were assigned as true positive. Particles (n = 304 for WT, n = 567 for chi2-sfGFP) were used for in situ subtomogram averaging.<br><br>cell type quantification via cryoET:<br>All high-quality tomograms were used for quantification of cells and cell types. Sample size for each dataset is stated in the figure, figure legend (n) or in the text.<br><br>light microscopy quantification<br>All high-quality light-microscopy images were used for quantification of cells and cell types. Sample size for each dataset is stated in the figure or figure legend (n).<br><br>intoxication assay: 12 larvae per treatment were used and three independent bioassays were undertaken. |

| Data exclusions | For all other experiments, no sample size determination was performed. |
| --- | --- |
| | Particles which were not fully inside the field of view were excluded. YenTc or YenTc-Chi2-sfGFP seen in tomograms of bad quality were excluded. Overlapping filament structures and filaments seen in tomograms of bad quality were excluded. |
| | Template matching: particles which were visually not identified as YenTc like density, such as ice chunks, membranes etc., were assigned as false positive and were excluded. |
| | For quantification in cryoET data, only cells with more than half of its size in the field of view were taken into account. |
| Replication | Replication of cryoET findings was not attempted as it is not applicable. |
| | Replications of light microscopy findings were successful at all attempts. All light microscopy findings were replicated at least 3 times. |
| | Replications of the intoxication assay were successful at all 3 attempts. |
| | All western blots and SDS-PAGEs were replicated at least 3 times with similar results. |
| Randomization | Extracted particles (for subtomogram averaging) were randomly assigned to two separate groups to calculate half-maps and goldstandard FSC. |
| | Light microscopy data was acquired at random locations. |
| | For other experiments, no randomization was performed. |
| Blinding | Particles identified via template matching were blindly distinguished (without knowledge of their cellular localization) between true positive (YenTc like density) and false positive particles (e.g. membrane, ice chunks). |
| | For other experiments, blinding was not attempted as it was not feasible. |

# Reporting for specific materials, systems and methods

We require information from authors about some types of materials, experimental systems and methods used in many studies. Here, indicate whether each material, system or method listed is relevant to your study. If you are not sure if a list item applies to your research, read the appropriate section before selecting a response.

## Materials & experimental systems

| n/a | Involved in the study |
| --- | --- |
| ☐ | ☒ Antibodies |
| ☒ | ☐ Eukaryotic cell lines |
| ☒ | ☐ Palaeontology and archaeology |
| ☐ | ☒ Animals and other organisms |
| ☒ | ☐ Clinical data |
| ☒ | ☐ Dual use research of concern |
| ☒ | ☐ Plants |

## Methods

| n/a | Involved in the study |
| --- | --- |
| ☒ | ☐ ChIP-seq |
| ☒ | ☐ Flow cytometry |
| ☒ | ☐ MRI-based neuroimaging |

## Antibodies

| Antibodies used | Membranes were incubated with 1 ug ml-1 polyclonal rabbit anti-YenA1 antibody (GenScript), 1:7,500 anti-GFP antibody (ab183734, Abcam), 1:1,000 anti-RecA (ab63797, Abcam) and 1:5000 horseradish peroxidase-conjugated secondary goat anti-rabbit IgG (31460, Invitrogen). |
| --- | --- |
| Validation | Validation of antibodies was done by manufacturer (SDS-PAGE of Antigen and Western Blot, Elisa) as well as with western blotting against Yersinia entomophaga wild-type, chi2-sfGFP, lysis cassette deficient and lysis casette deficient/YenTc deletion mutant. |

## Animals and other research organisms

Policy information about studies involving animals; ARRIVE guidelines recommended for reporting animal research, and Sex and Gender in Research

| Laboratory animals | Did not involve laboratory animals. |
| --- | --- |
| Wild animals | Costelytra giveni larvae. Field-collected 3rd instar insects were maintained until required in soil stored at 4C. |
| | Insects were killed by freezing for 48 hours prior to autoclaving. |
| Reporting on sex | n/a |
| Field-collected samples | Did not involve samples collected in the field. |
| Ethics oversight | No ethics approval was required. |

Note that full information on the approval of the study protocol must also be provided in the manuscript.

