## [Peer Review File · Nature Microbiology]

Peer Review Information

Journal: Nature Microbiology

Manuscript Title: Stepwise assembly and release of Tc toxins from Yersinia entomophaga

Corresponding author name(s): Professor Martin Pilhofer

Reviewer Comments & Decisions:Decision Letter, initial version:

Message: 26th September 2023

Dear Martin,

Thank you for your patience while your manuscript "Assembly and release of a Tc toxin" was under peer-review at Nature Microbiology. It has now been seen by 3 referees, whose expertise and comments you will find at the end of this email. Although they find your work of some potential interest, they have raised a number of concerns that will need to be addressed before we can consider publication of the work in Nature Microbiology.

In particular, referee #1's main concern is that the study lacks the application of a YenTc:sfGFP fusion (or other means) to support conclusions about the presence of YenTc. This referee also asks to provide a map of the TC gene cluster depicting all genes mentioned in this study. Referee #2 requests a more balanced discussion of which conclusions are based on experimental data and which ones represent more of hypotheses. Referee #3 feels that a more in-depth discussion on the potential role of the filaments in the context of YenTc assembly and release could be helpful. This referee also says that it would be interesting to see a comparative analysis of the YenTc toxin with other Tc toxins. Furthermore, referee #3 has some technical questions regarding the structural determination.

Should further experimental data allow you to address these criticisms, we would be happy to look at a revised manuscript.

Please include a data availability statement as a separate section after Methods but before references, under the heading "Data Availability". This section should inform readers about the availability of the data used to support the conclusions of your study. This information includes accession codes to public repositories (data banks for protein, DNA or RNA sequences, microarray, proteomics data etc...), references to source data published alongside the paper, unique identifiers such as URLs to data repository entries, or data set DOIs, and any other statement about data availability. At a minimum, you should include the following statement: "The data that support the findings of this study are available from the

3corresponding author upon request", mentioning any restrictions on availability. If DOIs are provided, we also strongly encourage including these in the Reference list (authors, title, publisher (repository name), identifier, year). For more guidance on how to write this section please see:

<http://www.nature.com/authors/policies/data/data-availability-statements-data-citations.pdf>

* If you have not done so already we suggest that you begin to revise your manuscript so that it conforms to our Article format instructions at <http://www.nature.com/nmicrobiol/info/final-submission>. Refer also to any guidelines provided in this letter.

When submitting the revised version of your manuscript, please pay close attention to our [href="https://www.nature.com/nature-portfolio/editorial-policies/image-integrity">Digital Image Integrity Guidelines.](https://www.nature.com/nature-portfolio/editorial-policies/image-integrity) and to the following points below:

Note: This url links to your confidential homepage and associated information about manuscripts you may have submitted or be reviewing for us. If you wish to forward this e-mail to co-authors, please delete this link to your homepage first.

Nature Microbiology is committed to improving transparency in authorship. As part of our efforts in this direction, we are now requesting that all authors identified as 'corresponding author' on published papers create and link their Open Researcher and Contributor Identifier (ORCID) with their account on the Manuscript Tracking System (MTS), prior to acceptance. This applies to primary research papers only. ORCID helps the scientific community achieve unambiguous attribution of all scholarly contributions. You can create and link your ORCID from the home page of the MTS by clicking on 'Modify my Springer Nature account'. For more information please visit please visit <a

<http://www.springernature.com/orcid>>www.springernature.com/orcid.

If you wish to submit a suitably revised manuscript we would hope to receive it within 3 months. If you cannot send it within this time, please let us know. We will be happy to consider your revision, even if a similar study has been accepted for publication at Nature Microbiology or published elsewhere (up to a maximum of 6 months).

Yours sincerely,

Reviewer Expertise:

Referee #1: Virulence of Yersinia species, Tc toxins, type 10 secretion systems

Referee #2: Tc toxins, insect pathogens

Referee #3: Structural Biology, cryo-EM, cryo-ET

Reviewer Comments:

Reviewer #1 (Remarks to the Author):

In their manuscript entitled "Assembly and release of a Tc toxin" by Pilhofer, Hurst, and colleagues (manuscript number NMICROBIOL-23071734A), the authors describe in detail the assembly and release of the Tc toxin produced by Yersinia entomophaga.

This is a highly interesting and clearly written manuscript easy to follow. It is experimentally sound and provides a huge set of very interesting data that shed further light on the molecular mechanism underlying the novel T10SS. All conclusions are very well supported by the experimental outcomes. The discussion is precisely written. In my view, this is a ground breaking paper in the field.

However, there are some points that need to be carefully addressed in a major revision. My most important concern is that the study lacks the application of a YenTc:sfGFP fusion (or other means) to support conclusions about the presence YenTc.

Major comments:

1. To ease understanding by readers not familiar with this topic, please provide a map of the TC gene cluster depicting all genes mentioned in this study.
2. Abstract line 19 and introduction line 45: I do not agree with the statement that the insecticidal toxin complex is a virulence factor directed against insects, plants, and humans. There is a huge body of literature that the TC acts against invertebrates and exhibits a proven insecticidal and nematocidal activity. This reviewer is aware of the (sporadic) literature that TC are active against mice or human cells. However, it is a widespread misunderstanding of microbial pathogenicity is that the mere presence of a toxin in a human pathogen allows the conclusion that this toxin plays a role in the interaction with its mammalian host. Rather, many researchers in the field assume that a huge spectrum of virulence factors is involved in colonisation and killing of non-vertebrates. References 3-7 do not support the generalisations made by the authors.
3. The Song et al. paper cited in line 47 is an in silico paper and mainly identified homologues of the enzymatically active TcC subunits. This does not exclude the presence of complete Tc clusters, however, I strongly suggest to weaken conclusions drawn from these data.

4. Lines 83-85: Please briefly rationalize why the Chi2-C-terminus is a good candidate. To my understanding, this a fusion between whole Chi2 and sfGFP, please clarify. Would you please mind to mention the molecular weight of Chi2 in the text? Please explain why you did not construct a YenTc-sfGFP fusion instead or in addition to Chi2-sfGFP.
5. Lines 109-110, fusion protein detection at poles, enlarged cells, lysing cells: this is an important and, to my view, correct observation. However, given that you observed hundreds of fluorescent cells, I would like to see photographs with more than only one or two examples for each category. Concentrate cells, or apply artificial overexpressing conditions?
6. Figure 2, YenTc-like densities: what is the evidence for interpreting these structures as YenTc proteins? Lines 133-134, this observation in my view does not clearly confirm the presence of YenTc (only the presence of these orange structures in lysing cells). Same for conclusion of Fig. 3e data: YenTc-like structures is not the same like YenTc. Did the authors accordingly investigate a YenTc deletion mutant as a negative control? This issue again raises the question why this study did not use a YenTc-sfGFP fusion, or at least antibodies against YenTc.
7. Line 144: 32% of all cells show signs of lysis, but less than 10% of cells show Chi2-sfGFP-based fluorescence. Does this mean that ~20% of lysing cells do not produce Chi2? Any explanation for this observation? There seems to be a significant difference in YenTc and Chi2 expression. Thus, this is another example for the urgent need to corroborate the results of the whole study by a YenTc-sfGFP fusion.
8. Lines 168-170: Without going into much detail with respect to the role of function of holin and endolysins with respect to virulence, I do not consider the Clostridium (and Salmonella literature as well) as highly interesting, but for several reasons (other toxins, other gene cluster, holin and endolysin not clustered but scattered over the chromosome) as a good introduction to the following. Hence, please cite publications from the Fuchs lab that, at least to my understanding, showed the functional activity of a similar holin/endolysin cassette (including the spanins) with respect to Tc release in vitro and in vivo in a strain (*Yersinia enterocolitica*) closely related to *Y. entomophaga*.
9. Lines 346-354: This is a very interesting hypothesis. To my understanding, the key sentence is the last one suggesting that M66/StcE is directly or indirectly linked to YenTc assembly, but the sentence starts with "Future studies will test...". To strengthen this part, I propose to state the hypothesis earlier in this paragraph, and to conclude like "Future studies will be performed to test this hypothesis...".
10. Lines 180-182 and 292-293: The authors are encouraged to bundle these two aspects, e.g. to maximize the benefit of lysing cells, their volume increases to allow the release of as much virulence factors as possible.
11. Line 175: the LC deletion mutant lacks *holA*, *pepB*, and *rz*, but not *rz1*, is this correct? Please explain why you left *rz1*, and if this fact weakens your interpretation of the data obtained with this mutant.
12. The title is rather short and not very instructive. Assembly and release of the insecticidal toxin complex by cell lysis of *Yersinia entomophaga*?

Minor comments:

1. I initially misunderstood the term filament as an external structure. I therefore propose to introduce this term in line 130 as "cytoplasmic filaments". In addition, please more clearly depict the filament bundles in the frame of Fig. 2f. An arrow indicating the filament in Fig. 3b is missing?
2. Please use a consistent style in references (avoid capital letters in title).
3. The authors should consider to show extended Figure 2 as a main figure or part thereof, the graphs on diseased animals is an important prerequisite for the following.
4. Line 173: I propose to cite the Schoof et al. publication (no. 29) at the end of this sentence.

5. Lines 186-187: Maybe I have overseen this, but you might include this observation in your assembly discussion (lines 289-329).
6. Line 215: Genes not in brackets.
7. Lines 315, 321: should read extended Fig. 6?

Reviewer #2 (Remarks to the Author):

This is a great paper and the authors should be congratulated. Its clearly been rather a mystery as to how such as large ABC toxin is assembled and released and the finding that this is done so by dedicating a set of 'suicide' cells to undergo phage-like lysis is important and should have broad implications for other bacterial toxins.

I really only have minor comments which the authors may (or may not) think will improve clarity.

1. I don't find the 'Yen'Tc label useful? Why not just use Tc, ABC toxin or Tc holotoxin. Given the extensive range of different names for the Tc's in a range of different bacteria I think we need to keep the nomenclature as simple as possible (for example we don't call them PhotoTc's). So why not stick with the nomenclature used in the title- just a Tc.
2. Clearly there is another paper detailing something similar from a different lab in the works- how do the authors propose to deal with this? Should they add a paragraph to the discussion to this effect if the paper is due to be published before this one?
3. 'bright' is rather qualitative- remove and leave as just 'signals were detected'.
4. The fact that these cells are stuffed with other virulence factors seems rather important to me and is perhaps somewhat under-stated? (As then it becomes a more general secretion mechanism for a whole cocktail of toxins/virulence factors). So it's tempting to go down the 'Suicide cells concentrate and release bacteria toxins including the Tcs' route? In that respect the first subheading of the Discussion may make a better title? 'Phage-like lysis releases a diverse array of toxins from a subset of bacterial cells'.
5. I'm not totally convinced by the stepwise nature of the process? Presumably the steps are inferred post hoc (down a microscope)? Can you stage these guys so that only one 'step' is present? Do some mutants leave the cells at one step only? So it's probably worth being more honest about the inferences here? (even if they are likely correct).
6. The protease filaments are intriguing- but there is no real data (unless I missed it?) presented on the 'either protease or Tc hypothesis'? How many times was this observed and how was it quantified? If this is really true then of course its not 'Suicide cells releasing a diverse array' but actually 'Suicide cells flip-flopping between the secretion of alternative toxins' (harder to get across as a simple high impact message- I realise but important).
7. Given the above question marks it would be useful to have a 'Conclusion' clarifying what is fact (data backed up quantitatively and with reps) versus what is assumption/sexy hypothesis/nice story- so the reader knows what might be done next to pin down the array of interesting questions raised- as happens with any great novel study.

Good luck with this- its a great paper- Richard ffrench-Constant

Reviewer #3 (Remarks to the Author):

Tc toxins are virulence factors found in various bacterial pathogens. The structure and action mechanism of Tc toxins are well-understood, but the assembly and release process remains unclear. Feldmüller et al use a range of experimental techniques including cryo-electron tomography, fluorescence light microscopy and proteomics to study this using Tc toxin from insect pathogen *Yersinia entomophaga* (YenTc). They also identify and characterize a phage-like lysis cassette as a key factor in this process. A stepwise mechanism for assembly/release is proposed, involving the priming of cells with virulence factors before their release by cell lysis.

Points:

- Overall the paper is well written and uses a variety of techniques with interesting new insight into Tc assembly and release.

- The paper is similar/overlapping with a study uploaded to biorxiv earlier in 2023 (<https://www.biorxiv.org/content/10.1101/2023.02.22.529496v1.full>). While not itself a problem, it might be helpful for the reader to refer to that also depending what the journal policy is on citing preprints. Of note, the volcano plots for the proteomics data are very similar.

- The identification of M66/StcE as the protein forming the observed filaments is interesting. A more in-depth discussion on the potential role of these filaments in the context of YenTc assembly and release could be helpful eg comparing to release mechanisms (eg T2SS) in other bacteria.

- It would be interesting to see a comparative analysis of the YenTc toxin with other Tc toxins, highlighting the similarities and differences in their assembly and release mechanisms.

- For the structural determination of purified YenTc, why was SPA not used instead of cryoET?

In Fig 1 – Given there is high resolution information for several of these components, can they be overlaid on the subtomogram? it would be helpful to further indicate overall dimensions, membrane spanning region.

In Fig 5 – Other than the 1-24 region denoted in the legend, are all other residues reliably predicted/present in the filament model?

In EDFig 1, the black arrowhead referred to in the legend denoting the presence of the GFP density is not present.

Author Rebuttal to Initial comments

Reviewer comments in Black.

Response to reviewers in Blue.

We appreciate the time invested by the reviewers and we thank them for their constructive comments, which helped to improve the manuscript.

Reviewer Comments:

Reviewer #1 (Remarks to the Author):

In their manuscript entitled "Assembly and release of a Tc toxin" by Pilhofer, Hurst, and colleagues (manuscript number NMICROBIOL-23071734A), the authors describe in detail the assembly and release of the Tc toxin produced by *Yersinia entomophaga*.

This is a highly interesting and clearly written manuscript easy to follow. It is experimentally sound and provides a huge set of very interesting data that shed further light on the molecular mechanism underlying the novel T10SS. All conclusions are very well supported by the experimental outcomes. The discussion is precisely written. In my view, this is a ground breaking paper in the field. However, there are some points that need to be carefully addressed in a major revision. My most important concern is that the study lacks the application of a YenTc:sfGFP fusion (or other means) to support conclusions about the presence YenTc.

We appreciate the enthusiastic comments by the reviewer and address the concern below.

Major comments:

1. To ease understanding by readers not familiar with this topic, please provide a map of the TC gene cluster depicting all genes mentioned in this study.

Thanks for this suggestion. We now show the map in Extended Figure 1 a.

2. Abstract line 19 and introduction line 45: I do not agree with the statement that the insecticidal toxin complex is a virulence factor directed against insects, plants, and humans. There is a huge body of literature that the TC acts against invertebrates and exhibits a proven insecticidal and nematocidal activity. This reviewer is aware of the (sporadic) literature that TC are active against mice or human cells. However, it is a widespread misunderstanding of microbial pathogenicity is that the mere presence of a toxin in a human pathogen allows the conclusion that this toxin plays a role in the interaction with its mammalian host. Rather, many researchers in the field assume that a huge spectrum of virulence factors is involved in colonisation and killing of non-vertebrates. References 3-7 do not support the generalisations made by the authors.

We agree with the reviewer and revised the abstract accordingly. The introduction states only that Tc genes are present in different pathogens but not that they that they play a role in pathogenicity against mammalian hosts.

3. The Song et al. paper cited in line 47 is an in silico paper and mainly identified homologues of the

enzymatically active TcC subunits. This does not exclude the presence of complete Tc clusters, however, I strongly suggest to weaken conclusions drawn from these data.

The text was revised accordingly (lines 45-51).

4. Lines 83-85: Please briefly rationalize why the Chi2-C-terminus is a good candidate. To my understanding, this a fusion between whole Chi2 and sfGFP, please clarify.

Chi2 is an integral part of YenTc (see Extended Fig. 1 a, and also shown in the structure in reference Piper et al., 2019) and we rationalized that placing a peripherally localized tag may not interfere with holotoxin formation and the ability to bind to target cells. We therefore fused sfGFP with the C-terminus of Chi2 using an 'Ala3Gly3' linker. This rationale is now stated in the text (lines 85-89).

Would you please mind to mention the molecular weight of Chi2 in the text?

The molecular weight (~69.7 kDa) is stated in the text (line 88).

Please explain why you did not construct a YenTc-sfGFP fusion instead or in addition to Chi2-sfGFP.

We do consider our construct being a YenTc-sfGFP fusion, since Chi2 is an integral part of the YenTc complex. This is shown in Extended Fig. 1 a, and it is also shown in the YenTc structure in reference Piper et al. 2019 (see also below Figure R1 a).

5. Lines 109-110, fusion protein detection at poles, enlarged cells, lysing cells: this is an important and, to my view, correct observation. However, given that you observed hundreds of fluorescent cells, I would like to see photographs with more than only one or two examples for each category. Concentrate cells, or apply artificial overexpressing conditions?

As suggested, additional examples are now shown in the new Extended Figure 4.

6. Figure 2, YenTc-like densities: what is the evidence for interpreting these structures as YenTc proteins?

Lines 133-134, this observation in my view does not clearly confirm the presence of YenTc (only the presence of these orange structures in lysing cells). Same for conclusion of Fig. 3e data: YenTc-like structures is not the same like YenTc. Did the authors accordingly investigate a YenTc deletion mutant as a negative control? This issue again raises the question why this study did not use a YenTc-sfGFP fusion, or at least antibodies against YenTc.

We followed the suggestion of the reviewer and also imaged lysed cells of a Δ YenTc mutant. As expected, no YenTc-like densities were observed, supporting our claims. This new data is now shown in Extended Figure 7.

That said, further strong evidence for the identity of YenTc-like densities is the following: In individual tomograms (e.g. Fig. 2 i), the structures resemble the overall architecture of the known YenTc structure from reference Piper et al. 2019, which is shown in the Figure R1 below (panel a).

Furthermore, YenTc was purified, identified by mass spectrometry (e.g. Extended Fig. 2 a),

imaged and subtomogram averages were generated. These subtomogram averages (see below Figure R1 b/d) show high similarity to the published structure (Fig. R1 a). Finally, we also identified YenTc-like densities in cellular tomograms in an unbiased way by template matching. Averaging of these particles again resulted in structures (Fig. R1 c/e) with high similarities to the published structure of YenTc (Fig. R1 a).Figure R1: Comparison of the published YenTc composite structure with subtomogram averages from the present study. (a) from Piper et al. 2019. (b) from Fig. 1 a. (c) from Fig. 2 h. (d) from Extended Fig. 1 f. (e) from Fig. 3 g.

7. Line 144: 32% of all cells show signs of lysis, but less than 10% of cells show Chi2-sfGFP- based fluorescence.

The reviewer correctly points out a discrepancy between LM (10% lysed cells) and cryoET (32% lysed cells). This difference is likely due to sample preparation, which leads to an overrepresentation of lysed cells that were imaged by cryoET. For cryoET, cells are pelleted before freezing, which may shift the fraction of lysed vs. intact cells. Importantly, the fraction of lysed cells observed by cryoET is comparable for wildtype and GFP-positive YenTc-sfGFP strains.

Does this mean that ~20% of lysing cells do not produce Chi2?

This statement is incorrect. Fig. 3 e shows that 98% of lysing cells contain YenTc (with Chi2 as an integral part). The subtomogram averages in fact do show an extra density that corresponds to Chi2 as a part of YenTc (see Fig. 3 g inset and also the structure in reference Piper et al., 2019).

Any explanation for this observation? There seems to be a significant difference in YenTc and Chi2 expression. Thus, this is another example for the urgent need to corroborate the results of the whole study by a YenTc-sfGFP fusion.

We believe this statement was due to a misunderstanding of the reviewer, to whom it was probably unclear that Chi2 is an integral component of YenTc (Fig. R1 a). We hope this has now been clarified.

8. Lines 168-170: Without going into much detail with respect to the role of function of holin and endolysins with respect to virulence, I do not consider the Clostridium (and Salmonella literature as well) as highly interesting, but for several reasons (other toxins, other gene cluster, holin and endolysin

not clustered but scattered over the chromosome) as a good introduction to the following. Hence, please cite publications from the Fuchs lab that, at least to my understanding, showed the functional activity of a similar holin/endolysin cassette (including the spanins) with respect to Tc release in vitro and in vivo in a strain (*Yersinia enterocolitica*) closely related to *Y. entomophaga*.

Revised as suggested by the reviewer.

9. Lines 346-354: This is a very interesting hypothesis. To my understanding, the key sentence is the last one suggesting that M66/StcE is directly or indirectly linked to YenTc assembly, but the sentence starts with "Future studies will test...". To strengthen this part, I propose to state

the hypothesis earlier in this paragraph, and to conclude like “Future studies will be performed to test this hypothesis...”.

This discussion has been revised according to the reviewer’s suggestion (lines 352-366), but also in light of additional new experiments generating and analyzing a $\Delta\text{LC}\Delta\text{M66}$ double mutant strain. Please see our detailed comments in response to reviewers #2 and #3 below.

10. Lines 180-182 and 292-293: The authors are encouraged to bundle these two aspects, e.g. to maximize the benefit of lysing cells, their volume increases to allow the release of as much virulence factors as possible.

Revised as suggested by the reviewer (lines 318-320).

11. Line 175: the LC deletion mutant lacks *holA*, *pepB*, and *rz*, but not *rz1*, is this correct? Please explain why you left *rz1*, and if this fact weakens your interpretation of the data obtained with this mutant.

We apologize that the statement in the text was incorrect, and it has now been revised (lines 181-182). In the generated mutant, *rz1* was not intact but rather partially deleted. The mutant contained only 127 base pairs of the 3'-end (compared to a total of 282 base pairs in the wildtype *rz1*). The interpretation of the data obtained with this mutant therefore holds up.

12. The title is rather short and not very instructive. Assembly and release of the insecticidal toxin complex by cell lysis of *Yersinia entomophaga*?

Thank you for this suggestion. We rather prefer to state these specifics in the abstract.

Minor comments:

1. I initially misunderstood the term filament as an external structure. I therefore propose to introduce this term in line 130 as “cytoplasmic filaments”. In addition, please more clearly depict the filament bundles in the frame of Fig. 2f. An arrow indicating the filament in Fig. 3b is missing?

All three suggestions have been implemented in the revised version.

2. Please use a consistent style in references (avoid capital letters in title).

Revised.

3. The authors should consider to show extended Figure 2 as a main figure or part thereof, the graphs on diseased animals is an important prerequisite for the following.

We now show the quantification of the assay in main Fig. 1 b.

4. Line 173: I propose to cite the Schoof et al. publication (no. 29) at the end of this sentence.

Revised accordingly.

5. Lines 186-187: Maybe I have overseen this, but you might include this observation in your assembly discussion (lines 289-329).

Revised accordingly.

6. Line 215: Genes not in brackets.

Revised.

7. Lines 315, 321: should read extended Fig. 6?

Revised accordingly.

Reviewer #2 (Remarks to the Author):

This is a great paper and the authors should be congratulated. Its clearly been rather a mystery as to how such as large ABC toxin is assembled and released and the finding that this is done so by dedicating a set of 'suicide' cells to undergo phage-like lysis is important and should have broad implications for other bacterial toxins.

Thanks for your constructive comments. We do share the enthusiasm for the project.

I really only have minor comments which the authors may (or may not) think will improve clarity.

1. I don't find the 'YenTc' label useful? Why not just use Tc, ABC toxin or Tc holotoxin. Given the extensive range of different names for the Tc's in a range of different bacteria I think we need to keep the nomenclature as simple as possible (for example we don't call them PhotoTc's). So why not stick with the nomenclature used in the title- just a Tc.

While the authors share the reasoning for a simple nomenclature, the term "YenTc" has been widely used throughout the literature. We therefore prefer to stick with this term.

2. Clearly there is another paper detailing something similar from a different lab in the works- how do the authors propose to deal with this? Should they add a paragraph to the discussion to this effect if the paper is due to be published before this one?

We are aware of the preprint from the Raunser Lab and now comment on the study (lines 67-69) and cite it in the text. The insights from this study are generally consistent with our data, with differences in the cellular ultrastructure of lysing cells and the absence of M66 filaments. Since a peer-reviewed version of the Sitsel et al. study has not been published, we prefer to omit a detailed discussion in our manuscript.

3. 'bright' is rather qualitative- remove and leave as just 'signals were detected'.

Revised accordingly.

4. The fact that these cells are stuffed with other virulence factors seems rather important to me and is perhaps somewhat under-stated? (As then it becomes a more general secretion mechanism for a

whole cocktail of toxins/virulence factors). So it's tempting to go down the 'Suicide cells concentrate and release bacteria toxins including the Tcs' route? In that respect the first subheading of the Discussion may make a better title? 'Phage-like lysis releases a diverse array of toxins from a subset of bacterial cells'.

As suggested, we revised the text to emphasize the priming of cells with several other virulence factors (e.g. line 23 [abstract]). We prefer to keep the original title.

5. I'm not totally convinced by the stepwise nature of the process? Presumably the steps are inferred post hoc (down a microscope)? Can you stage these guys so that only one 'step' is present? Do some mutants leave the cells at one step only? So it's probably worth being more

honest about the inferences here? (even if they are likely correct).

Our Δ LC mutant generates a population of cells, in which ~83% are primed with virulence factors but the cells cannot proceed towards YenTc assembly and release. We consider this mutant as a way of “staging” the cells in an initial step and these cells already proved useful for investigating e.g. filaments or assembly upon triggering lysis externally. It will be very interesting to identify means to also stage cells prior to the final step of release in the future, which may be catalyzed by spanins.

6. The protease filaments are intriguing- but there is no real data (unless I missed it?) presented on the 'either protease or Tc hypothesis'?

In order to address the comment of this reviewer, we generated a Δ LC Δ M66 double mutant. In this mutant, the majority of cells were stuck in the primed state, however, without the expression of M66 filaments. These data are shown in the new Extended Fig. 14. We used this mutant to test whether the absence of M66 affected the assembly of YenTc holotoxin. A purification from these cells identified abundant YenTc particles with similar architecture. These data are shown in the new Extended Fig. 14 and discussed in the revised text (lines 352-362). We speculate that M66 is not required for general YenTc assembly, even though we cannot exclude differences whose detection would require high-resolution structure determination. Finally, we also set out to test a generated Δ M66 mutant for its efficiency to kill grass grubs. Unfortunately, at the time of writing, grubs were off-season and could not be obtained from the environment to conduct the intoxication experiments.

How many times was this observed and how was it quantified?

A quantification of cells that show presence of filaments (by visual inspection of tomograms) can be found in lines 166-169 (for Δ LC), in Fig. 2 j (for *chi2-sfGFP*), Fig. 3 e (for WT).

If this is really true then of course its not 'Suicide cells releasing a diverse array' but actually 'Suicide cells flip-flopping between the secretion of alternative toxins' (harder to get across as a simple high impact message- I realise but important).

The reviewer raises an interesting point. Our observation that 98% of cells with signs of cell lysis showed YenTc densities (see Fig. 3 e) suggests that almost the entire population of primed cells expresses YenTc. We also show that in intact cells, Chi2-sfGFP expression always correlates with the abundance of filaments. It is therefore unlikely that there are distinct subpopulations expressing either YenTc or M66. That said, our present dataset does not allow us to assess the expression of other virulence factors on a subpopulation level.

7. Given the above question marks it would be useful to have a 'Conclusion' clarifying what is fact (data backed up quantitatively and with reps) versus what is assumption/sexy hypothesis/nice story- so the reader knows what might be done next to pin down the array of interesting questions raised-

as happens with any great novel study.

Throughout the discussion, we tried to combine concluding statements with references to the respective figures/tables, showing the original data and quantifications. The discussion was also revised to take the analysis of the new $\Delta LC\Delta M66$ double mutant strain into consideration. This allowed us, for instance, to better distinguish between the alternative hypotheses that were raised under point 6 by this reviewer.

Good luck with this- its a great paper- Richard ffrench-Constant

Reviewer #3 (Remarks to the Author):

Tc toxins are virulence factors found in various bacterial pathogens. The structure and action mechanism of Tc toxins are well-understood, but the assembly and release process remains unclear. Feldmüller et al use a range of experimental techniques including cryo-electron tomography, fluorescence light microscopy and proteomics to study this using Tc toxin from insect pathogen *Yersinia entomophaga* (YenTc). They also identify and characterize a phage-like lysis cassette as a key factor in this process. A stepwise mechanism for assembly/release is proposed, involving the priming of cells with virulence factors before their release by cell lysis.

Points:

- Overall the paper is well written and uses a variety of techniques with interesting new insight into Tc assembly and release.

We are glad that the reviewer finds our contributions interesting.

- The paper is similar/overlapping with a study uploaded to biorxiv earlier in 2023 (<https://www.biorxiv.org/content/10.1101/2023.02.22.529496v1.full>). While not itself a problem, it might be helpful for the reader to refer to that also depending what the journal policy is on citing preprints. Of note, the volcano plots for the proteomics data are very similar.

As suggested by reviewers #2 and #3, we now comment on the preprint (lines 67-69) and cite it in the text.

- The identification of M66/StcE as the protein forming the observed filaments is interesting. A more in-depth discussion on the potential role of these filaments in the context of YenTc assembly and release could be helpful eg comparing to release mechanisms (eg T2SS) in other bacteria.

In order to address the open question of M66 function, we generated a $\Delta\text{LC}\Delta\text{M66}$ mutant. In this mutant, the majority of cells were stuck in the primed state, however, without the expression of M66 filaments. These data are shown in the new Extended Fig. 14. We used this mutant to test whether the absence of M66 would affect the assembly of YenTc holotoxin. A purification from these cells identified abundant YenTc particles with similar architecture. These data are shown in the new Extended Fig. 14 and discussed in the revised text (lines 352-362). We speculate that M66 is not required for general YenTc assembly, even though we cannot exclude differences whose detection would require high-resolution structure determination. Finally, we also set out to test a ΔM66 mutant for its efficiency to kill grass grubs. Unfortunately, at the time of writing, grubs were off-season and could not be obtained from the environment to conduct the intoxication experiments.

- It would be interesting to see a comparative analysis of the YenTc toxin with other Tc toxins, highlighting the similarities and differences in their assembly and release mechanisms.

The conservation of lysis cassette gene clusters in Tc-positive organisms was shown previously and suggests a conserved release mechanism (see Palmer et al., 2021 and Schoof et al., 2023 and Sitsel et al., 2023). This is also discussed in the text (lines 170-180, 279-284, 285-289). Furthermore, Sanger et al., 2022 supports this notion for *Y. enterocolitica*, where a lysis cassette was shown to be required for Tc release.

Engineering a tagged and functionally active Tc complex in other organisms will allow for more detailed comparisons but this is clearly outside the scope of this paper. We would like to note

that the presence of the additional Chi2 YenTc component presented us with a suitable system for tagging, which is likely much more difficult in other systems.

- For the structural determination of purified YenTc, why was SPA not used instead of cryoET?

The goal of the experiment was to determine whether sfGFP could be seen as an extra density, in comparison to the already published high-resolution structure of wildtype YenTcA. For this purpose, a low-resolution average was sufficient. For this proof of concept, it was most straightforward for us to use the established tomography workflow.

In Fig 1 – Given there is high resolution information for several of these components, can they be overlaid on the subtomogram? it would be helpful to further indicate overall dimensions, membrane spanning region.

To indicate the fit of the overall dimensions, we show the published YenTc composite model in Figure R2 (a) below, as well as our subtomogram averages docked with the published YenA high resolution structure (Fig. R2 b/c). Densities that are unaccounted for represent YenBC (as we used five-fold symmetrized subtomogram averaging which would not allow to fit the crystal structure), Chi1 and parts of YenA/Chi2 that were not resolved in previous structures.

Figure R2: Comparison of the published YenTc composite structure with subtomogram averages from the present study. (a) from Piper et al. 2019. (b) side and top views of our YenTc wildtype subtomogram average (mesh) docked with the YenA structure from Piper et al. 2019 (red). (c) side and top views of our YenTc Chi2-sfGFP subtomogram average (mesh) docked with the YenA structure from Piper et al. 2019 (red). Bar: 10 nm.In Fig 5 – Other than the 1-24 region denoted in the legend, are all other residues reliably predicted/present in the filament model?

The cryoEM densities did not leave any residues unaccounted for, when fitting the AlphaFold2 model (except for residues 1-24). The predicted structure (AF-A0A3S6EYX4-F1) also has no major features outside the boundaries of the density map. The AlphaFold2 “per residue confidence score” (pLDDT) for the structure was generally very high with values of >90 (except for residues 1-24).

In EDFig 1, the black arrowhead referred to in the legend denoting the presence of the GFP density is not present.

Sorry - we now inserted the arrowhead.

Decision Letter, first revision:

Message: Our ref: NMICROBIOL-23071734B

20th December 2023

Dear Martin,

Thank you for submitting your revised manuscript "Assembly and release of a Tc toxin" (NMICROBIOL-23071734B). It has now been seen by the original referees and their comments are below. The reviewers find that the paper has improved in revision, and therefore we'll be happy in principle to publish it in Nature Microbiology, pending minor revisions to satisfy the referees' final requests and to comply with our editorial and formatting guidelines.

We are now performing detailed checks on your paper and will send you a checklist detailing our editorial and formatting requirements in about two weeks. Please do not upload the final materials and make any revisions until you receive this additional information from us.

Thank you again for your interest in Nature Microbiology. Please do not hesitate to contact me if you have any questions.

Sincerely,

Reviewer #1 (Remarks to the Author):

1The authors revised their manuscript "Assembly and release of a Tc toxin" by Pilhofer, Hurst, and colleagues (manuscript number NMICROBIOL-23071734A) that describes in detail the assembly and release of the Tc toxin produced by *Yersinia entomophaga*. Additional comment: The authors correctly cite a complementary study by Sitsel et al. Lines 279-281 should read "in exoproteome release of *Y. entomophaga* (22) and in the insecticidal pathogenicity of *Y. enterocolitica* (30)". More importantly, I miss a discussion of the Sitsel et al. paper. Such a discussion would strengthen both papers.

Major comments:

Comment 1: Would you please mind to add a map of the phage cassette of *Y. entomophaga* MH96 (YeRER) as shown in Fig. 1A in Schoof et al. *Microbiol. Spectrum* 2023 to Fig. 1a?

Comments 2 and 3: I strongly propose to rewrite lines 44-45 and name it as it is: the Tc toxins are insecticidal toxins, and homologous genes were identified in pathogens of insects, plants, and humans. However, toxicity or activity has been clearly demonstrated so far only against nematodes and insect larvae, but not towards plants or humans.

Comment 4: I am not fully convinced, in particular with respect to the argument that the tag does not interfere with holotoxin formation. Please provide experimental evidence from your data and photos, and cite literature that support your assumption.

Comment 5: satisfyingly answered.

Comment 6: I suggest to briefly summarize all your arguments as partially done in lines 150-153.

Comment 7: Please briefly address the discrepancy between LM and cryET data in the results.

Comment 8: Please apologize my incorrect grammar. The toxin release by *Clostridium* and *Salmonella* is highly interesting, but to my opinion not closely related to the *Yersinia* LC with respect to mechanisms and regulation, and therefore does not fit well to this paragraph. I propose to eliminate the sentence on *Clostridium* here (lines 173-174). It might be mentioned elsewhere to discuss alternative holin/endolysin associated mechanisms.

Moreover, I suggest to rewrite lines 176-180 like that: "We recently identified...a lysis cassette encoding a holing etc., which are located outside the Tc gene cluster...However, we recently demonstrated that this LC is required for general exoprotein release". This is a stronger statement in my view.

Comment 9: satisfyingly answered.

Comment 10: satisfyingly answered.

Comment 11: satisfyingly answered.

Comment 12: I would like to leave the decision on a more appropriate title to the editor.

Minor comments: satisfyingly answered.

Reviewer #2 (Remarks to the Author):

No further comments.

Reviewer #3 (Remarks to the Author):

The authors have addressed my queries in their revised manuscript.

Final Decision Letter:

Message: 17th January 2024

Dear Martin,

I am pleased to accept your Article "Stepwise assembly and release of Tc toxins from *Yersinia entomophaga*" for publication in Nature Microbiology. Thank you for having chosen to submit your work to us and many congratulations.

After the grant of rights is completed, you will receive a link to your electronic proof via

3email with a request to make any corrections within 48 hours. If, when you receive your proof, you cannot meet this deadline, please inform us at rjsproduction@springernature.com immediately. You will not receive your proofs until the publishing agreement has been received through our system

Please note that *Nature Microbiology* is a Transformative Journal (TJ). Authors may publish their research with us through the traditional subscription access route or make their paper immediately open access through payment of an article-processing charge (APC). Authors will not be required to make a final decision about access to their article until it has been accepted. [Find out more about Transformative Journals](https://www.springernature.com/gp/open-research/transformative-journals)

Authors may need to take specific actions to achieve [compliance](https://www.springernature.com/gp/open-research/funding/policy-compliance-faqs) with funder and institutional open access mandates. If your research is supported by a funder that requires immediate open access (e.g. according to [Plan S principles](https://www.springernature.com/gp/open-research/plan-s-compliance)) then you should select the gold OA route, and we will direct you to the compliant route where possible. For authors selecting the subscription publication route, the journal's standard licensing terms will need to be accepted, including [self-archiving policies](https://www.nature.com/nature-portfolio/editorial-policies/self-archiving-and-license-to-publish). Those licensing terms will supersede any other terms that the author or any third party may assert apply to any version of the manuscript.

We welcome the submission of potential cover material (including a short caption of around 40 words) related to your manuscript; suggestions should be sent to Nature Microbiology as electronic files (the image should be 300 dpi at 210 x 297 mm in either TIFF or JPEG format). Please note that such pictures should be selected more for their aesthetic appeal

than for their scientific content, and that colour images work better than black and white or grayscale images. Please do not try to design a cover with the Nature Microbiology logo etc., and please do not submit composites of images related to your work. I am sure you will understand that we cannot make any promise as to whether any of your suggestions might be selected for the cover of the journal.

Congratulations once again and I look forward to seeing the article published.

With kind regards,